# Lightweight Online Adaption for Time Series Foundation Model Forecasts

**Thomas L. Lee** [* 1 2]   **William Toner** [* 3]   **Rajkarn Singh** [3]   **Artjom Joosen** [3]   **Martin Asenov** [3]

## Abstract

Foundation models (FMs) have emerged as a promising approach for time series forecasting. While effective, FMs typically remain fixed during deployment due to the high computational costs of learning them online. Consequently, deployed FMs fail to adapt their forecasts to current data characteristics, despite the availability of online feedback from newly arriving data. This raises the question of whether FM performance can be enhanced by the *efficient* usage of this feedback. We propose *ELF* to answer this question. ELF is a lightweight mechanism for the online adaption of FM forecasts in response to online feedback. ELF consists of two parts: **a)** the *ELF-Forecaster* which is used to learn the current data distribution; and **b)** the *ELF-Weighter* which is used to combine the forecasts of the FM and the ELF-Forecaster. We evaluate the performance of ELF in conjunction with several recent FMs across a suite of standard time series datasets. In *all* of our experiments we find that using ELF improves performance. This work demonstrates how efficient usage of online feedback can be used to improve FM forecasts.

## 1. Introduction

Over the last decade there has been rapid development in machine learning models for time series forecasting (Darlow et al., 2023; Lim & Zohren, 2021; Nie et al., 2022). This has prompted practitioners in fields requiring accurate forecasting to adopt and deploy increasingly advanced models to remain competitive. Although these deep models perform well, training them can be prohibitive, requiring substantial

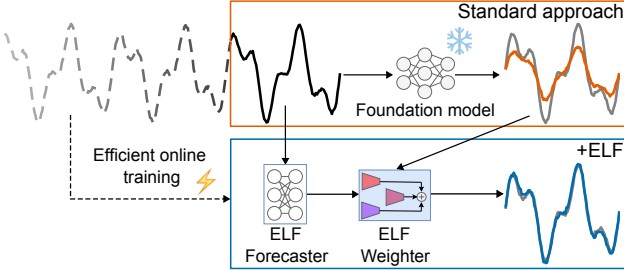

*Figure 1.* **Exploitation of online information for forecast adaption in deployment:** Time series foundation models (FMs) are typically seen as fixed when deployed. We propose a method, *ELF*, to efficiently improve their forecasts online by leveraging the feedback available in realistic deployment scenarios. ELF consists of two components: the *ELF-Forecaster*, a lightweight forecaster, learnt online; and the *ELF-Weighter* a dynamic method to adapt the FM forecasts by combining it with the ELF-Forecaster's forecasts.

computational resources, machine learning expertise, and ample data (Mercier et al., 2021). These practical hurdles have spurred interest in foundation models (FMs) for time series forecasting (Miller et al., 2024; Ansari et al., 2024), which diverge from the standard paradigm of training on a specific dataset. Instead, FMs are trained across multiple large datasets (Woo et al., 2024), enabling them to generalise to new, unseen time series and deliver strong *zero-shot* forecasting performance (Ansari et al., 2024; Rasul et al., 2023). This avoids retraining a new model for each task, allowing FMs to be used out-of-the-box even with no available training data, ML expertise, or computational resources.

While effective, time series FMs remain fixed during deployment, making them non-adaptive to shifts in the underlying distribution. In time series deployment scenarios however, new data arrives continuously, providing feedback on forecast accuracy and changes in the underlying behaviour of the time series (Zhang et al., 2024). This raises the possibility of using online feedback to enhance FM performance. However, this can be challenging as any online adaptation mechanism must be computational efficient to be practical in real-world time series settings (Joosen et al., 2024). Potential naive approaches to online adaption, such as regular retraining or continual finetuning on available data, are at present prohibitively expensive—as demonstrated in Section 5.3 and Verwimp et al. (2023). Currently, there is *no*

[*]Equal contribution [1]School of Informatics, University of Edinburgh, UK [2]Work done while an intern at Huawei [3]Huawei SIR Lab, Edinburgh Research Centre, UK. Correspondence to: Thomas L. Lee <T.L.Lee-1@sms.ed.ac.uk>, William Toner <william.toner2@huawei.com>.

*Proceedings of the $42^{nd}$ International Conference on Machine Learning*, Vancouver, Canada. PMLR 267, 2025. Copyright 2025 by the author(s).

*general, efficient method for leveraging online feedback in time series FMs at deployment.*

In this work, we propose *ELF*, an efficient approach for online adaption of FM forecasts at deployment. ELF does not alter the parameters of an FM, which remain fixed during deployment, instead adapting the *forecasts* output by the FM. In effect, we take a non-online FM and turn it into an efficient online model, FM+*ELF* (see Figure 1). This approach is lightweight, running on a single CPU, making implementation cheap. Importantly, ELF can be used *out-of-the-box*—i.e., without human supervision—to enhance the performance of any FM.

ELF consists of two components: the *ELF-Forecaster*, a lightweight forecast model which is trained online, and the *ELF-Weighter* which combines the forecasts of the FM and the ELF-Forecaster using an online weighting policy, effectively ensembling the two models. To ensure that ELF is computationally efficient we design a complex linear model to use as the ELF-forecaster, exploiting the Woodbury matrix identity to fit it efficiently online. To construct the ELF-weighter we build upon exponential weighting, an approach with a longstanding pedigree in the online learning community (Cesa-Bianchi & Lugosi, 2006). We construct a weighter made out of a fast adapting component to extract local information and a slow component to extract global information. This allows the weighter to quickly adapt to distribution shift. We experimentally demonstrate consistent performance improvements when using ELF, across multiple state-of-the-art foundation models and across the standard time-series benchmark datasets.

Our work consists of three main contributions:

1. We propose *ELF*, an efficient way to exploit the online feedback available in the deployment stage of a time series foundation model to improve performance.

2. Design a lightweight linear forecaster applied in the Fourier domain—the *ELF-Forecaster*—which can be efficiently fit online via the Woodbury matrix identity.

3. Design a dynamic weighter—the *ELF-Weighter*—to combine the ELF-Forecaster and FM forecasts. This is constructed out of fast and slow components to quickly adapt the weighting to shifting data distributions.

## 2. Related Work

Currently, the main paradigm to construct (zero-shot) time series FMs is to collect a large pretraining set of time series data and then use it to learn an LLM-like transformer model (Rasul et al., 2023; Chen et al., 2024; Liang et al., 2024). For example, *Chronos* (Ansari et al., 2024), *Moirai* (Woo et al., 2024) and *TimesFM* (Das et al., 2024) are all time series FMs which are trained on large curated pretrain-ing sets of time series, consisting of billions of training points, and whose backbones consist of LLM-based transformer architectures—such as Chronos, which uses the T5 architecture (Roberts et al., 2019). Also, recently there have been new time series FMs which have gone against this trend by not using LLM architectures as their backbones. For instance, *TTM* (Ekambaram et al., 2024) uses the TSMixer architecture (Ekambaram et al., 2023) as its backbone, which is specific to time series, resulting in a much smaller model size when compared to methods using LLM backbones. While, *VisionTS* (Chen et al., 2024) uses a masked autoencoder (He et al., 2022) as its backbone and does not use time series data for pretraining, instead using images from the ImageNet dataset.

A major focus of this work is the exploitation of online feedback available at the deployment stage of a time series forecaster. The idea of leveraging online feedback in deployment to improve performance of an ML system has a long history (Hoi et al., 2021; Polikar et al., 2001; Bottou & Cun, 2003). Currently, this concept falls within the domain of continual or lifelong learning for deep learning methods (De Lange et al., 2021; Wang et al., 2024). Some of these works, for instance, investigate how to update a model online given the newly available data, which is either from the same distribution or from a shifting distribution (Aljundi et al., 2019; Lee & Storkey, 2024b). Importantly, much of the research on continual learning has focused on vision or text tasks (Qu et al., 2021; Wu et al., 2024), with comparatively little attention given to the time series domain (Besnard & Ragot, 2024). The studies that have explored continual learning for time series forecasting concentrate on methods for updating the weights of deep learning models (Ao & Fayek, 2023; Pham et al., 2022; Zhang et al., 2024). This has two problems in the context of time series: **a)** updating the weights of a deep learning model, especially of the size of a FM, at the frequency often required for time series data and with the resources usually available (e.g. CPUs) can often make such methods infeasible to use in practice (Diao et al., 2024; Ekambaram et al., 2024). And **b)** online updating of deep models suffers from the problems of catastrophic forgetting and plasticity loss (Kirkpatrick et al., 2017; Dohare et al., 2023; De Lange et al., 2021). Solutions to this currently require retraining on large amounts of historic data and complex, model-specific learning routines (Yang et al., 2024). This is in contrast to the focus of our work, which looks at the efficient online adaption of FM forecasts, so that it can be widely used in the real world.

## 3. Rolling Window Forecasting

When deploying a time series forecast model in practice, new data becomes available over time. To model this realistic scenario, practitioners typically adopt a *rolling window*

approach in which the time series is processed one time step at a time (Nie et al., 2022). At each time step, a *context window* containing the previous $L$ time-series values is constructed, from which the model produces a forecast for the next $H$ steps. $H$ is referred to as the *forecast horizon*. By progressing through the time series step-by-step, previous model forecasts can be evaluated against the actual ground truth values of the forecast that subsequently occur, called the *target*. This *online feedback* of past forecasts can be used to improve future forecasts.

In this work we look at the rolling window setting, where the parameters of ELF are altered every $M$ time steps using online feedback; a process we refer to as *updating*. We keep track of each time ELF is updated by using a counter referred to as the *update step* which we denote by $\tau$, to differentiate it from our notation for time step $t$. For every $M$ time steps $t \mapsto t + M$, there is a single update step $\tau \mapsto \tau + 1$.

### 3.1. Notation

We list the notation used in this paper below. For notational simplicity, *here and in Section 4, we only describe the univariate case.* This is when each value in the time series is a scalar. This simplification is without loss of generality, as ELF forecasts each dimension of the time series, called *channels*, separately—i.e. it is channel independent.

- $L$: Context length (number of time steps in the input sequence).

- $H$: Forecast horizon (number of future time steps to predict).

- $c$: Number of channels (distinct time series).

- $\boldsymbol{x}$: Context window, $\boldsymbol{x} \in \mathbb{R}^L$.

- $M$: The number of time steps between subsequent updating of ELF.

- $t$: Current time step.

- $\tau$: Update Step, number of times forecaster has been updated using online feedback. Given by $\lfloor \frac{t}{M} \rfloor$.

- $\boldsymbol{y}, \hat{\boldsymbol{y}}$: Target and forecast respectively, $\boldsymbol{y}, \hat{\boldsymbol{y}} \in \mathbb{R}^H$.

- $X, Y$: *Design* matrices containing all observed context and target vectors respectively up to the current time step. $\tilde{X}, \tilde{Y}$ Fourier domain representations of the design matrices.

## 4. ELF: Online Adaption of Forecasts

We propose a method for **E**nsembling with online **L**inear **F**orecaster (**ELF**), to improve the performance of time series FMs during deployment. **ELF** is a lightweight approach consisting of two parts:

1. The *ELF-Forecaster*: a lightweight forecast model, trained online on the most recently observed time series data.

2. The *ELF-Weighter*: an online weighting mechanism which adjusts the forecasts of the FM by combining it with the forecasts of the ELF-Forecaster, dynamically ensembling the forecasters.

We display the components of ELF in Figure 2 and describe them in turn below.

### 4.1. ELF-Forecaster

The ELF-Forecaster is a linear forecast model used to provide forecasts based on the data seen online from the time series. We use a linear forecaster for three reasons: **a)** they have good performance in time series forecasting (Zeng et al., 2023); **b)** as required by our setting, they can be efficiently updated online; and **c)** online updating of linear models does not suffer from the same problems of catastrophic forgetting as the online updating of neural networks (De Lange et al., 2021). In this section we first describe how the ELF-Forecaster generates a forecast and then how it is fit efficiently online. Algorithmic descriptions of these processes are presented in Appendix A.1.3.

**Forecasting** The linear ELF-Forecaster forecasts in the Fourier domain, being inspired by the FITS forecaster presented in Xu et al. (2023) (as discussed in Appendix A.1.2). To produce a forecast the ELF-Forecaster takes in a context vector $\boldsymbol{x} \in \mathbb{R}^L$ and applies the Discrete Fourier Transform (DFT)

$$\mathrm{DFT}(\boldsymbol{x})_k := \frac{1}{\sqrt{L}} \sum_{n=0}^{L-1} x_n e^{\frac{-2i\pi kn}{L}}. \qquad (1)$$

After this, to improve updating efficiency, the highest frequencies are discarded. Concretely, given $\alpha \in [0, 1]$ we crop out those components $\mathrm{DFT}(\boldsymbol{x})_k$ for which $k$ satisfies $\left(\frac{\alpha}{2}\right) < \frac{k}{L} < \left(1 - \frac{\alpha}{2}\right)$. Removing these frequency components reduces the dimensionality from $L$ to $2\lfloor \frac{\alpha L}{2} \rfloor \approx \alpha L$. We multiply the resulting low-dimensional vector by a complex weight matrix $W$ of shape $\alpha L \times \frac{\alpha H}{2}$ ($\boldsymbol{x} \mapsto \boldsymbol{x}^T W$). The output of this operation is zero-padded up to dimensionality $(1 + \frac{H}{2})$. Finally, we apply the inverse Real Fourier Transform (iRFT) to generate the forecast $\hat{\boldsymbol{y}} \in \mathbb{R}^H$.[1] The only parameters of the ELF-Forecaster is the complex weight matrix $W$.

**Online Fitting** The ELF-Forecaster is refit once every *update step* $\tau$ (every $M$ time steps). We fit the ELF-Forecaster by optimizing the mean squared error (MSE) between its forecasts and the ground truth across all past data up to

---

[1]The RFT of vector $\boldsymbol{x}$ is obtained by taking the first $L/2 + 1$ components of the $\mathrm{DFT}(\boldsymbol{x})$.

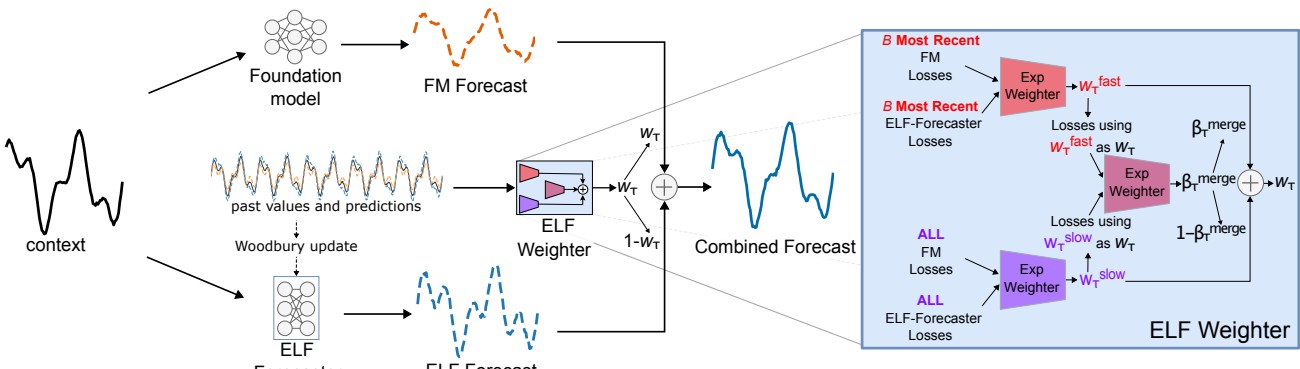

*Figure 2.* **Forecast construction for FM+*ELF*:** The left side of the figure shows how ELF adjusts the forecast of the FM. This is achieved by combining the FM forecast with the forecasts of the online (Woodbury) updated ELF-Forecaster using a weighted average. The weights $w_\tau$ in the weighted average are determined by the ELF-Weighter based on previous performance of the FM and ELF-Forecaster. The schematic in the blue box to the right, zooms into the components of the ELF-Weighter: the fast, slow and merge weighters. Where the fast weighter gives a weighting based on recent performance; the slow weighter gives a weighting based on performance across the whole history. The merge weighter is used to combine the weights given by the fast and slow weighters. Each weighter uses exponential weighting to construct their weights; the difference being the losses used as input. While the slow weigher can be seen as using all previous losses to compute its weight, we note that only the last $B$ losses need to stored by ELF.

the current time step $t = \tau M$. Importantly, since the ELF-Forecaster is a linear model it can be fit in closed-form, avoiding the need for gradient-based optimisation (Toner & Darlow, 2024), and without storing all the data. We note here that we initialise $W$ to be the weights of a naive seasonal forecaster. Then, when there is enough data ($\tau M \geq L + H$), we fit the $W$ for the first time.

To refit the ELF-forecaster at each update step $\tau$, we proceed as follows. Let $(x_1, x_2, \ldots, x_{\tau M})$ be the values of a univariate time series up to the current time step $\tau M$. Let $\boldsymbol{x}_s := (x_s, x_{s+1}, \ldots, x_{s+L-1})$, $\boldsymbol{y}_s := (x_{s+L}, x_{s+L+1}, \ldots, x_{s+L+H-1})$ denote the context and its respective target starting at time step $s$. Let $X \in \mathbb{R}^{N \times L}$ and $Y \in \mathbb{R}^{N \times H}$, denote matrices formed from all contexts and targets up to the current time step so that the $i^{\text{th}}$ row of $X$ is $\boldsymbol{x}_i$. (Note that $N := \tau M - L - H$; this choice of $N$ ensures we do not fit on future time steps). Now, let $\tilde{X}$ denote the matrix obtained by taking the DFT of each row of $X$ and let $\tilde{Y}$ denote the matrix obtained by applying the RFT to each row of $\tilde{Y}$. The (L2-regularised) Ordinary Least-Squares (OLS) solution for the complex weight matrix of the ELF-Forecaster can be expressed as

$$W := (\tilde{X}^*\tilde{X} + \lambda I)^{-1}\tilde{X}^*\tilde{Y}, \qquad (2)$$

where $\lambda$ is the regularisation coefficient and $*$ denotes the conjugate transpose.

In a naive implementation of the above one could store all observed data and recompute Equation 2 every update step. However, this approach is expensive. Instead of storing $\tilde{Y}$ we can store $\tilde{X}^*\tilde{Y}$. This speeds up refitting as we can

efficiently update this matrix at each update step $\tau$ via

$$\tilde{X}^*\tilde{Y} \mapsto \tilde{X}^*\tilde{Y} + \tilde{X}_\tau^*\tilde{Y}_\tau.$$

where $\tilde{X}_\tau, \tilde{Y}_\tau$ denote the matrices formed from the $M$ most recently observed contexts/targets respectively. We additionally speed up refitting in another two ways: **a)** storing and incrementally updating the matrix $(\tilde{X}^T\tilde{X} + \lambda I)^{-1}$ using the *Woodbury Matrix Identity* (Woodbury, 1950). This, in conjunction with storing $\tilde{X}^*\tilde{Y}$, can be used to update $W$ using Equation 2, without needing to store all previously observed data. **b)** Removing high frequency components of $\tilde{X}, \tilde{Y}$, that is removing some specific columns of the matrices. We describe both approaches below:

**The Woodbury Matrix Identity** is a method for efficiently recomputing the inverse of a matrix after a low-rank update (Woodbury, 1950). We apply this in our setting to enable efficient recomputation of $(\tilde{X}^*\tilde{X} + \lambda I)^{-1}$ upon receiving new data. Specifically, letting

$$A^{-1} := (\tilde{X}^*\tilde{X} + \lambda I)^{-1},$$

the Woodbury update is defined as

$$A^{-1} \mapsto A^{-1} - B,$$

where

$$B := A^{-1}\tilde{X}_\tau^*(I + \tilde{X}_\tau A^{-1}\tilde{X}_\tau^*)^{-1}\tilde{X}_\tau A^{-1}$$

If model refitting occurs regularly (so that $M < L$) this approach is more efficient than the alternative of storing/updating $(\tilde{X}^*\tilde{X} + \lambda I)$ and recomputing the inverse each update step. This is because rather than inverting a

large matrix of dimension $L \times L$ (as in Equation 2), we invert a smaller matrix of dimension $M \times M$.

**Removing High Frequencies** Fitting the ELF-Forecaster in Fourier space provides a convenient mechanism for dimensionality reduction of the context and target vectors via the removal of high frequency components. This allows us to reduce the number of model parameters and further speed up fitting. Specifically, given some $\alpha \in [0, 1]$ (typically $\alpha = 0.9$) and the Fourier domain representation of a context (or target) vector, we retain the lowest $\alpha$ proportion of frequencies, discarding the rest. In practice this means removing some of the columns of the matrices denoted $\tilde{X}, \tilde{Y}$ in Eqn 2. This filtering technique, reduces the dimensionality of the weight vector being learned from $L \times H$ to $(\alpha L \times \frac{\alpha H}{2})$, speeding up the fitting, with minimal impact on performance as demonstrated in our ablations (see Appendix C.8).

## 4.2. ELF-Weighter

The *ELF-Weighter* combines the forecasts of the FM and the ELF-Forecaster; adapting the FM's forecast by incorporating the knowledge learnt online by the ELF-Forecaster. We combine the forecasts by taking a weighted average of them. In other words, we construct a dynamic weighted ensemble of the FM and ELF-forecaster. Denoting the forecasts of the FM and the ELF-Forecaster at time step $t$ by $\hat{\boldsymbol{y}}_{t,FM}$ and $\hat{\boldsymbol{y}}_{t,EF}$, respectively, the combined forecast is given by

$$\hat{\boldsymbol{y}}_{t,ELF} = w_\tau \hat{\boldsymbol{y}}_{t,FM} + (1 - w_\tau)\hat{\boldsymbol{y}}_{t,EF}. \tag{3}$$

The weight $w_\tau \in [0, 1]$ is adjusted online by the ELF-Weighter to reflect the changes in the relative performance between the FM's and ELF-Forecaster's forecasts. We use a weighted average to combine forecasts because the general idea has been shown theoretically to improve performance (e.g., Theorem 4.1). We also note that the simpler approach of using an unweighted average ($w_\tau = 0.5$) leads to poor performance, as demonstrated in Appendix C.10.

The basic building block of the ELF-Weighter is exponential weighting which is a celebrated method from the online learning community (Cesa-Bianchi & Lugosi, 2006; Rakhlin & Kleiner, 2008) and can be used to weight forecasts. However, the standard way to perform exponential weighting has the drawback of being quite slow to adapt to changes in the relative performance differences between the forecasters. To be more adaptive we are inspired by Complementary Learning System theory (McClelland et al., 1995; Kumaran et al., 2016; Ba et al., 2016) to use a combination of a slow exponential weighter, which learns weights based on the *whole* history, and a fast weighter, which only uses the *recent* history. Below, we first describe exponential weighting in general and then the fast and slow weighers used by ELF-Weighter to produce $w_\tau$.

**Exponential Weighting** is a method for combining $K$ forecasters by computing a weighted average of their forecasts, where the weights are learned online based on the past losses of each forecaster. We describe first the general case, where for $K$ forecasters $\omega_{\tau,k}$ denotes the weight for the $k^{\text{th}}$ forecaster at update step $\tau$. The weights are updated at update step $\tau$ by the learning rule:

$$\omega_{\tau,k} = \frac{\omega_{\tau-1,k} e^{-\eta \text{Loss}_{\tau,k}}}{\sum_{k'=1}^{K} \omega_{\tau-1,k'} e^{-\eta \text{Loss}_{\tau,k'}}},$$

where $\text{Loss}_{\tau,k}$ denotes the loss incurred by the $k^{\text{th}}$ forecaster on update step $\tau$, $\eta$ denotes a predefined learning rate and where we assume $\omega_{0,k} = 1$ for all $k$. By recursively expanding the update rule it is possible to give a closed form solution to the weights learnt at update step $\tau$,

$$\omega_{\tau,k} = \frac{e^{-\eta \sum_{\tau'=1}^{\tau} \text{Loss}_{\tau',k}}}{\sum_{k'=1}^{K} e^{-\eta \sum_{\tau'=1}^{\tau} \text{Loss}_{\tau',k'}}}.$$

Hence the weights learnt by exponential weighting is a softmax of the cumulative losses incurred so far by the forecasters. This update rule has two desirable properties: **a)** the worse a forecaster performs the smaller its weight, and **b)** the weighter is affected less and less by individual updates as time progresses. Additionally, there is a well known theoretical result which bounds the cumulative loss when using exponential weighting; theoretically motivating its use:

**Theorem 4.1.** *(Cesa-Bianchi & Lugosi, 2006; Rakhlin & Kleiner, 2008) Given a convex loss function, define the loss of the weighted-average forecaster at some update step $\tau$ as* $\text{Loss}_{\tau,weighted}$ *and define regret at time* $\text{T}$ *as*

$$R_T = \sum_{\tau'=1}^{\text{T}} \text{Loss}_{\tau',weighted} - \min_{k \in \{1,...,K\}} \sum_{\tau'=1}^{\text{T}} \text{Loss}_{\tau',k}.$$

*Then for a maximum incurred loss of $L_{\max}$ and a learning rate of $\eta = \frac{1}{L_{\max}} \sqrt{\frac{8 \ln K}{\text{T}}}$ we have that*

$$R_T \leq L_{\max} \sqrt{\frac{\text{T}}{2} \ln K}.$$

**Slow and Fast Weighters** While exponential weighting is an effective approach it has the drawback that it does not quickly adapt to distribution shift (Jadbabaie et al., 2015; Cesa-Bianchi et al., 2012; Zhao et al., 2020). This is because the losses incurred over the most recent time steps are given the same importance as those in the far past when constructing the weights. Hence, it takes several update steps for there to be enough losses from a new data distribution to outweigh older losses to correctly adjust the weights. To remedy this drawback we look at using a combination of two weighters (as shown in Figure 2): **a)** a **slow**

weighter which is an exponential weighter, between the FM and ELF-Forecaster. The update for *the slow weight for the FM forecast* $w_\tau^{slow}$ at update step $\tau$ is

$$w_\tau^{slow} = \frac{w_{\tau-1}^{slow} e^{-\eta \text{Loss}_{\tau,1}}}{w_{\tau-1}^{slow} e^{-\eta \text{Loss}_{\tau,1}} + (1 - w_{\tau-1}^{slow}) e^{-\eta \text{Loss}_{\tau,2}}}.$$

**b)** A **fast** weighter, identical to the slow weighter but only using losses from the last $B$ update steps. *The fast weight for the FM forecast* $w_\tau^{fast}$ at update step $\tau$ is

$$w_\tau^{fast} = \frac{e^{-\eta \sum_{\tau'=\tau-B}^{\tau} \text{Loss}_{\tau',1}}}{\sum_{k'=1}^{2} e^{-\eta \sum_{\tau'=\tau-B}^{\tau} \text{Loss}_{\tau',k'}}}.$$

By only using the last $B$ updates for the fast weighter we aim to make it more adaptive than the slow weighter to the current relative performance difference between the FM and ELF-Forecaster. Additionally, note that in our setting we always have $K = 2$, this means we effectively have a single weight for each weighter where, for instance, *the slow weight for the ELF-Forecaster is* $1 - w_\tau^{slow}$.

**Merging Fast and Slow Weights**  The final part of the ELF-Weighter is a mechanism for synthesising a single weight $w_\tau$ from the fast and slow weights ($w_\tau^{fast}, w_\tau^{slow}$ respectively). $w_\tau$ is used for combining the forecasts of the FM and ELF-Forecaster as in Equation 3. We use an exponential weighter, the **merge** weighter, to produce this final weight. Specifically, we record the losses $\text{Loss}_{\tau,f}$ obtained when combining the FM and ELF-Forecaster using the fast weight (i.e. setting $w_\tau := w_\tau^{fast}$) and the losses $\text{Loss}_{\tau,s}$ obtained when using slow weight ($w_\tau := w_\tau^{slow}$). We then use exponential weighting to fit a weight $\beta_\tau^{merge}$:

$$\beta_\tau^{merge} = \frac{\beta_{\tau-1}^{merge} e^{-\eta \text{Loss}_{\tau,f}}}{\beta_{\tau-1}^{merge} e^{-\eta \text{Loss}_{\tau,f}} + (1 - \beta_{\tau-1}^{merge}) e^{-\eta \text{Loss}_{\tau,s}}}.$$

This *merge* weight is then used to combine the fast and slow weights via

$$w_\tau = \beta_\tau^{merge} w_\tau^{fast} + (1 - \beta_\tau^{merge}) w_\tau^{slow},$$

to generate the final weight given in Equation 3. An algorithmic description of the ELF-Weighter is given in Appendix A.2.

## 5. Experiments

### 5.1. Experimental Setup

To evaluate the performance of ELF we simulate how it would perform in a deployment scenario. Specifically, we adopt the rolling window setting described in Section 3, moving through the time series and producing forecasts one time step at a time. We update ELF every $M = 200$ time steps using the online feedback provided by the newest 200 data points.

**Foundation Models (FMs)**  Our experiments explore using ELF in combination with several of the most recent and well know (zero-shot) FMs for time series: Chronos (tiny) (Ansari et al., 2024), TTM (revision 2) (Ekambaram et al., 2024), TimesFM (Das et al., 2024), Moirai (small) (Woo et al., 2024) and VisionTS (Chen et al., 2024). We compare the performance of each FM with and without using ELF.

**Datasets**  The datasets we evaluate on are ETTh1, ETTh2, ETTm1, ETTm2, Weather, Traffic, ECL, Solar and US Weather (given in Zhou et al. (2021); Wu et al. (2021); Liu et al. (2022); Darlow et al. (2024)). These are standard datasets used widely in time series work (Ekambaram et al., 2024) and importantly, none of the FMs used these datasets for pretraining. This is with the exception of TimesFM which uses Traffic and ECL for training, hence we do not report results for it for those two datasets. Furthermore, we look at these datasets for three different prediction horizons: $H = 30, 96, 336$, which were chosen to be comparable to previous work (Woo et al., 2024; Ekambaram et al., 2024). Throughout all experiments we use a context length of $L = 520$, which is a standard context length for the FMs used in our experiments (Ansari et al., 2024). Additional details about our experiments, including additional hyperparameter settings, are given in Appendix B.

**Metrics**  We evaluate our results using Mean Absolute Scaled Error (MASE) (Hyndman & Koehler, 2006). This is defined for a given forecast $\hat{y}$, context $x$ and target $y$ as

$$\text{MASE}(\hat{y}, y, x) = \frac{L - S}{H} \frac{\sum_{i=1}^{H} |\hat{y}_i - y_i|}{\sum_{i=1}^{L-S} |x_i - x_{i+S}|}$$

where $S$ represents the seasonality. MASE measures the absolute error between the forecast and target, normalised by the error of a naive seasonal forecaster on the context. This allows for comparisons across time series with varying scales over time. This is important in the rolling window setting, as it is unrealistic to assume that each channel is scaled to unit variance.

We also report results using the Root Mean Squared Scaled Error (RMSSE) in Appendix C.12. This metric is defined similarly to MASE but uses RMSE instead of MAE (Hyndman & Koehler, 2006). The conclusions drawn from both metrics are the same.

### 5.2. Main Results

Table 1 presents the results of our experiments applying ELF to FMs, showing that ELF improves performance in all cases. The table displays the MASEs of each FM and the difference in MASE when using the FM with ELF (the *+ELF* columns). For the difference, a negative number means that FM+*ELF* performs better than using the FM on its own, which we colour *green*. As shown by the table

*Table 1.* **MASE of time series foundation models with and without using *ELF*:** A lower MASE is better and we present results for each dataset over multiple forecast horizon lengths denoted as $H$ in the table. The results show that by using *ELF* we improve performance across all datasets and forecast lengths tested.

| Dataset | $H$ | TTM | +ELF(↓) | TimesFM | +ELF(↓) | VisionTS | +ELF(↓) | Chronos | +ELF(↓) | Moirai | +ELF(↓) |
|---|---|---|---|---|---|---|---|---|---|---|---|
| | | | | | | Time Series FMs | | | | | |
| ETTh1 | 30 | 0.930 | -0.019 | 0.913 | -0.022 | 0.967 | -0.063 | 0.936 | -0.047 | 1.010 | -0.089 |
| | 96 | 1.081 | -0.014 | 1.107 | -0.049 | 1.084 | -0.028 | 1.120 | -0.063 | 1.168 | -0.086 |
| | 336 | 1.286 | -0.006 | 1.350 | -0.062 | 1.306 | -0.022 | 1.361 | -0.074 | 1.417 | -0.101 |
| ETTh2 | 30 | 1.472 | -0.018 | 1.470 | -0.024 | 1.542 | -0.071 | 1.455 | -0.021 | 1.536 | -0.058 |
| | 96 | 2.786 | -0.016 | 2.799 | -0.040 | 2.787 | -0.029 | 2.801 | -0.047 | 2.863 | -0.062 |
| | 336 | 6.802 | -0.011 | 6.816 | -0.040 | 6.763 | -0.008 | 6.850 | -0.092 | 6.913 | -0.105 |
| ETTm1 | 30 | 0.802 | -0.048 | 0.833 | -0.079 | 1.021 | -0.252 | 0.891 | -0.139 | 1.078 | -0.311 |
| | 96 | 0.973 | -0.053 | 1.016 | -0.097 | 1.050 | -0.130 | 1.164 | -0.243 | 1.239 | -0.310 |
| | 336 | 1.205 | -0.063 | 1.276 | -0.131 | 1.216 | -0.077 | 1.489 | -0.341 | 1.479 | -0.327 |
| ETTm2 | 30 | 0.799 | -0.036 | 0.814 | -0.057 | 1.040 | -0.251 | 0.843 | -0.085 | 0.946 | -0.164 |
| | 96 | 0.991 | -0.038 | 1.029 | -0.077 | 1.088 | -0.124 | 1.107 | -0.150 | 1.151 | -0.180 |
| | 336 | 1.320 | -0.040 | 1.429 | -0.128 | 1.351 | -0.072 | 1.478 | -0.193 | 1.522 | -0.219 |
| US Weather | 30 | 0.893 | -0.036 | 0.868 | -0.041 | 1.027 | -0.172 | 0.939 | -0.091 | 0.896 | -0.068 |
| | 96 | 1.123 | -0.040 | 1.164 | -0.102 | 1.155 | -0.085 | 1.204 | -0.121 | 1.143 | -0.088 |
| | 336 | 1.296 | -0.044 | 1.364 | -0.123 | 1.286 | -0.048 | 1.423 | -0.163 | 1.334 | -0.111 |
| Weather | 30 | 0.887 | -0.033 | 0.798 | -0.020 | 1.342 | -0.449 | 1.077 | -0.230 | 1.161 | -0.261 |
| | 96 | 1.205 | -0.043 | 1.269 | -0.220 | 1.436 | -0.253 | 1.697 | -0.538 | 1.720 | -0.503 |
| | 336 | 1.576 | -0.040 | 3.084 | -1.591 | 1.691 | -0.140 | 2.099 | -0.571 | 2.073 | -0.494 |
| Solar | 30 | 1.091 | -0.031 | 1.097 | -0.058 | 1.004 | -0.020 | 0.984 | -0.030 | 1.222 | -0.132 |
| | 96 | 1.129 | -0.031 | 1.201 | -0.097 | 1.079 | -0.021 | 1.080 | -0.046 | 1.308 | -0.164 |
| | 336 | 1.166 | -0.033 | 1.248 | -0.100 | 1.236 | -0.087 | 1.107 | -0.028 | 1.292 | -0.119 |
| ECL | 30 | 1.003 | -0.086 | — | — | 0.982 | -0.107 | 0.874 | -0.034 | 1.225 | -0.288 |
| | 96 | 1.106 | -0.094 | — | — | 1.090 | -0.104 | 1.015 | -0.054 | 1.303 | -0.275 |
| | 336 | 1.279 | -0.086 | — | — | 1.368 | -0.177 | 1.236 | -0.079 | 1.476 | -0.264 |
| Traffic | 30 | 0.887 | -0.067 | — | — | 0.962 | -0.140 | 0.659 | -0.015 | 0.770 | -0.029 |
| | 96 | 0.920 | -0.077 | — | — | 0.943 | -0.112 | 0.746 | -0.033 | 0.752 | -0.017 |
| | 336 | 0.965 | -0.084 | — | — | 1.012 | -0.127 | 0.923 | -0.100 | 0.801 | -0.019 |

we improve the performance in all cases as we have green numbers for all FMs across all of the datasets looked at. In some cases the improvement is quite large. For instance, by using ELF to adapt the forecasts of VisionTS we get an average improvement of over $10\%$. These results illustrate that efficiently and effectively exploiting the online feedback in the rolling window setting allows ELF to improve the forecasts of FMs.

**5.3. Comparison to Continual Learning Methods**

While we have shown that ELF improves the forecasts of FMs, it is also important to see how well it compares to other methods which use online feedback to improve perfor-

mance. To do this, we benchmark ELF against four different continual learning methods: **a)** TAFAS (Kim et al., 2025), which, like ELF, is a method to adapt the forecasts of an FM using online feedback. TAFAS differs to ELF in the fact that it adjusts the context not only the forecast, its module to adjust forecasts is conditioned on the FM forecasts not context and it uses gating (Chen et al., 2018) not exponential weighting to adjust the strength of the forecast adaption. **b)** FSNet (Pham et al., 2023), which is a modified temporal convolutional network (Bai et al., 2018), such that it has improved performance when trained continually. **c)** OneNet (Zhang et al., 2023), which is an ensemble of two forecasters fit online, where one forecaster leverages cross-channel information and the other cross-time information.

*Table 2.* **Results of time series online adaption methods:** We report for each method the rel. MASE w.r.t the naive seasonal forecaster, averaged over the forecast horizon lengths $\{30, 96, 336\}$ and the seconds per update step, measured for the ETTh1 dataset with a prediction length of 96. We compute seconds per update step using 2 Intel(R) Xeon(R) Platinum 8168 CPUs and put in brackets the proportional speedup of updating using ELF versus the other online adaption methods. The results show that *ELF* performs the best in terms of both forecast accuracy (MASE) and compute efficiency (sec. per update step).

| Dataset | TTM+*ELF* | TTM+*TAFAS* | OneNet-TTM | OneNet | FSNet |
|---|---|---|---|---|---|
| ETTh1 | **0.882** | 0.893 | 1.084 | 1.123 | 1.204 |
| ETTh2 | **0.947** | 0.954 | 1.183 | 1.334 | 1.130 |
| ETTm1 | **0.810** | 0.843 | 1.177 | 1.146 | 1.235 |
| ETTm2 | **0.814** | 0.834 | 1.213 | 1.373 | 1.363 |
| US-Weather | **0.817** | 0.838 | 1.109 | 1.105 | 1.198 |
| Weather | **0.770** | 0.790 | 3.551 | 2.043 | 2.510 |
| Solar | **0.936** | 0.962 | 1.250 | 1.300 | 1.810 |
| ECL | **0.831** | 0.897 | 1.278 | 1.299 | 3.916 |
| Traffic | **0.676** | 0.737 | 0.756 | 0.803 | 1.124 |
| Sec. Per Update Step (ELF Speedup) | **0.38** | 3.76 (**10x**) | 30.66 (**81x**) | 43.12 (**113x**) | 23.69 (**62x**) |

The ensembled forecasters used are typically FSNet models, to mitigate the problems of continual learning. However, here we also look at using TTM as the cross-time model (OneNet-TTM) to explore a variant that is more like ELF. We note that all of these methods are fit with gradient decent which can be computationally costly and often leads to problems when continually learning (De Lange et al., 2021). Additionally, they require an offline training set to initialise their parameters, unlike ELF. Further details of these methods are presented in Appendix B. The results of using these methods in the rolling window setting are presented in Table 2, where we use TTM as the FM for TAFAS (results for other FMs are presented in Appendix C.2) and compare to TTM+*ELF*, as TTM is the best performing FM in our experiments. The results show that TTM+*ELF* performs better than all of the other methods across all datasets tested. This demonstrates the effectiveness of how ELF uses online feedback to adjust forecasts.

**Computational Cost** A core characteristic of ELF is that it is computationally inexpensive. For example, when deploying on two CPUs, ELF adds only an additional 0.38 seconds per update relative to employing the FM without online adaption. This speed is crucial since, during deployment, online updating must occur faster than new data arrives. To contextualise the speed of ELF, in Table 2 we show the computational cost of other online learning methods. We find that ELF is 10x faster than any other method. We also in Appendix C.3 run an experiment where we continually finetune TTM and find that ELF is 2506x faster. Also, ELF also outperforms this approach, achieving an average 5.89% gain in predictive performance on the ETT datasets. These results demonstrate the computational efficiency of ELF.

### 5.4. Why Does ELF Improve Performance?

While Table 1 demonstrates that ELF improves the performance of FMs, it is important to analyse why this is the case. We believe that ELF provides a benefit in two main ways: **a)** the ELF-Weighter can identify shifts in data distribution, ensuring that the combined forecast is well suited to the current features of the time series. **b)** By leveraging online feedback the ELF-Forecaster fits to the specific dynamic features of the time series, enabling the combined forecast to more accurately model these features. We evidence both points below.

The ELF-Weighter makes ELF sensitive to the changing data distribution of the time series. This is achieved by the ELF-Weighter quickly adjusting the weighting between the ELF-Forecaster and FM when distribution shifts occur. An example of the weights adjusting to a shift in distribution is shown in Figure 3, which depicts the evolution of the FM weights ($w_\tau$) for Chronos on ETTh1. Specifically, the weight given to the FM for the purple channel rapidly increases in the cream shaded region. Additionally, in Figure 4 we plot a moving average of the MASEs over time of the Moirai FM, the ELF-Forecaster and the combined Moirai+*ELF* during a snippet of the ETTh1 dataset. The Figure shows how the ELF-Weighter dynamically weights the forecasts of the ELF-forecaster and the FM based on their performance on the local data distribution. This allows ELF to generally generate forecasts which are superior to either forecaster individually. Notably, Figure 4 also shows that utilising the ELF usually provides a benefit regardless of whether, for the current data distribution, the FM outperforms the ELF-Forecaster (cream shaded regions) or vice versa (light-blue shaded regions). This all provides evidence

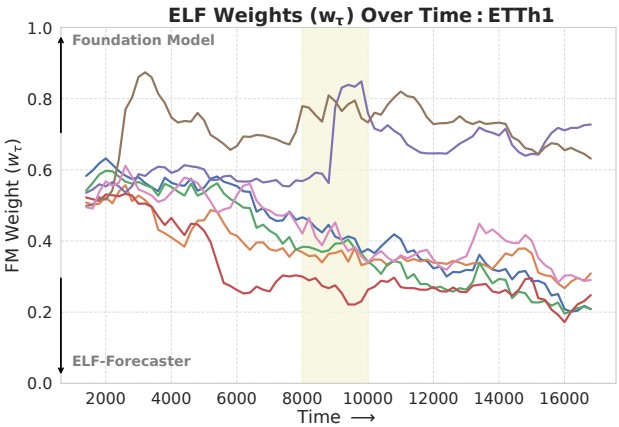

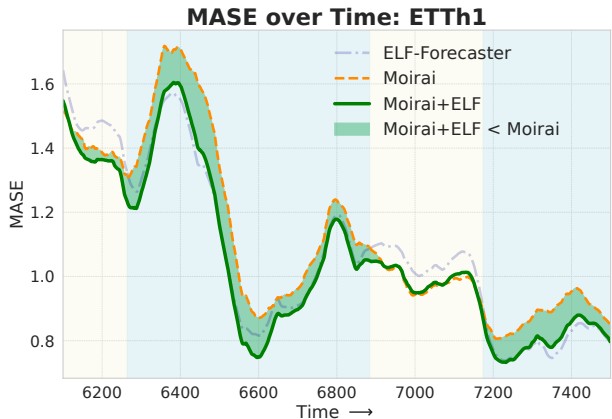

*Figure 3.* **ELF weight** ($w_\tau$) **over time for the Chronos FM for each channel of the ETTh1 dataset:** Each line in the plot shows the ELF weight for a channel. A weight of 1 means that the ELF-Forecaster has no contribution to the final forecast, similarly, a weight of 0 means that the FM is not contributing to the final forecast. The plot shows that for 5 out of 7 channels the FM weights gradually decrease; this reflects the fact that as the ELF-Forecaster observes more data its performance improves and it is upweighted by the ELF-Weighter. Occasionally, the weights adapt quickly to changes in the underlying time series, for example the purple channel shows a rapid weighting change between steps $8,000$-$10,000$ (shaded in cream).

*Figure 4.* **MASEs of the Moirai FM, Moirai+*ELF* and ELF-Forecaster over time on a snippet from channel 6 of the ETTh1 dataset:** The plot shows during certain intervals (shaded in cream) Moirai outperforms the ELF-Forecaster, on other occasions the reverse is true (shaded in light blue). While the ELF-Weighter ensures that the performance of the combined forecast (the green dashed line) generally outperforms either individual forecast. This demonstrates a reason for why using ELF improves performance, where the gain in using it is given by the green shaded region.

to the fact that ELF both adjusts to shifts in the data distribution and by doing so increases the accuracy of forecasts.

Figure 3 also shows that for most channels, the FM weight tends to decrease over time. This phenomenon suggests that as the ELF-Forecaster observes more dataset-specific data it generally becomes better fitted to the given time series and is consequently up-weighted. This has the follow-on effect that the FM's forecast can be increasingly adapted to the characteristics of the time series it has been deployed on. An additional discussion of how the ELF-Forecaster identifies time series specific features to improve performance is presented in Appendix C.13. We also remark that, there are some channels in Figure 3 where the FM consistently has a larger weight than the ELF-Forecaster (e.g. the brown line) and hence is performing better. This justifies the decision to make ELF channel-independent, as if were not, it would not be able to weight these channels differently from the others.

### 5.5. Ablations

To analyse the respective contribution of different parts of ELF we perform several ablations: **a)** in Appendix C.10 we look at the respective performance of the two main components of the ELF-Weighter, the fast and slow weighters. We find that using both leads to a gain compared to using either individually. **b)** In Appendices C.7, C.8 and C.9 we ablate the ELF-Forecaster. Demonstrating that it performs better

than replacing it with other types of forecasters, for example more complex forecasters like FSNet or forecasters that predict the residual of the FM forecast. Additionally, we establish that filtering high frequencies and using the Woodbury matrix identity leads to significant speed up while only having a negligible impact on forecasting performance. **c)** In Appendix C.6 we analyse the role of the updating frequency $M$. We observe that by updating ELF more frequently one may improve its forecasting performance albeit at the cost of compute performance.

## 6. Conclusions

In this work we look at how to efficiently improve the forecasts of a deployed time series foundation model (FM). We propose *ELF*, a method which exploits the fact in deployment there is online feedback on previous forecasts as new data arrives. ELF leverages this feedback in two ways: **a)** to train a lightweight linear forecaster (ELF-Forecaster) on the newly arriving data to learn the up-to-date data distribution; and **b)** to adapt the FMs forecasts by combining them with the forecasts generated by the ELF-Forecaster using a learnt weighting mechanism—the ELF-Weighter. We demonstrate experimentally that by using *ELF* we improve performance consistently across all datasets and FMs looked at. This indicates that exploiting the online feedback given in deployment is an effective way to boost the performance of FMs. Crucially, this online adaption of forecasts is achieved in a FM-agnostic manner and with a small enough overhead that it can be widely used in the real world.

## Acknowledgements

We would like to kindly thank Luke Darlow for his help and Thomas L. Lee would like to thank his PhD supervisor, Amos Storkey.

## Impact Statement

This paper looks at the general area of time series forecasting. While there are many potential societal consequences of work in time series forecasting, due to the general scope of our work we do not feel there are any specific societal consequences to note here.

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

# A. Additional Methodological Details for ELF

## A.1. Implementation Details of the ELF-Forecaster

**Initialisation** Initially there is no data to fit ELF-Forecaster since we assume that we are deploying zero-shot without access to a training dataset. Consequently, we are required to wait a small amount of time before the ELF-Forecaster can first be fit to data, at least $L + H$ time steps. To handle this period we initialise the linear model to the naive seasonal forecaster. Then after seeing at least $L + H$ time steps we fit the ELF-Forecaster on all the available data at the nest update step $\tau$.

**Instance Norm** The ELF-Forecaster utilises a variant of instance norm at inference time. Given a context vector $x$ and a target vector $y$, and a forecast model $f$, this involves normalising $x$ by its mean $\mu(x)$ applying a model $f$ on the normalised $x'$, and adding this mean back onto the prediction $\hat{y}$. Formally;

$$x' = x - \mu(x),$$
$$\hat{y} = f(x'),$$
$$\hat{y}_{\text{out}} = \hat{y} + \mu(x).$$

### A.1.1. Numerical Issues and Resolutions

The naive approach for handling the repeated refitting of the ELF-Forecaster is to store $X^T X$ and $X^T Y$. As one observes new contexts and targets these quantities can be updated via;

$$X^T X \mapsto X^T X + X_M^T X_M,$$
$$X^T Y \mapsto X^T Y + X_M^T Y_M.$$

After which one may recompute $(X^T X + \lambda I)^{-1} X^T Y$ to produce the updated weight matrix. While this approach is effective and requires only the storage of $L \times L$ and $L \times H$ matrices, one must take care that the cumulative sums $X^T X, X^T Y$ don't grow too large. For example, in the context of serverless computing practitioners are often interested in predicting function requests. The number of these requests can sometimes be in the millions per hour (Joosen et al., 2023; Diao et al., 2024). Consequently, in this setting, over the course of a year the values of $X^T X$ will grow to be in the order of $10^{17}$. Using, say, 32-bit float the large growth in the magnitude of the elements of $X^T X$ can result in a loss of numerical precision. To address this issue we adopt a subtly different approach, instead storing $\frac{X^T X}{N}, \frac{X^T Y}{N}$ where $N$ is the number of data instances which have been observed thus far. This ensures that the magnitude of the values in these matrices remains approximately constant. Specifically, given $M$ new data instances $X_M, Y_M$ one updates $\frac{X^T X}{N}$ in the following way:

$$\frac{X^T X}{N} \mapsto \frac{N}{N + M} \left( \frac{X^T X}{N} \right) + \frac{X_M^T X_M}{N + M}.$$

The update is similar for $\frac{X^T Y}{N}$.

We then observe that

$$(X^T X + \lambda I)^{-1} X^T Y = \left( \frac{X^T X}{N} + \frac{\lambda}{N} I \right)^{-1} \frac{X^T Y}{N}$$

Thus we can compute the OLS solution using the scaled quantities $\frac{X^T X}{N}$ and $\frac{X^T Y}{N}$ annealing the regularisation parameter $\lambda$ by scaling by $\frac{1}{N}$ as we see more data.

**Data Scaling** Typically, when training deep models on time series, practitioners adopt the approach of normalising the data. The standard approach is to compute the mean and the standard deviation of each series on some training set and then scale so that the data is zero mean and unit variance (Nie et al., 2022). The validation and test data are scaled using these same parameters derived from the training set. This technique handles the radically different scales between different time series, and mirrors data standardisation practices in other areas of machine learning such as computer vision. In an online setting however we assume that there is no training dataset from which these metrics can be computed. Thus we cannot adopt this approach. We resolve this by computing a running standard deviation for each channel which are updated online as more data is observed using Welford's online algorithm (Welford, 1962). These values are used to scale the data before fitting the linear model. Note that, at inference time, no data scaling is needed before applying the linear model since $\alpha A \left( \frac{x}{\alpha} \right) = A x$ for any $\alpha$.

### A.1.2. How Does The ELF-Forecaster Differ From FITS?

Our ELF-Forecaster architecture is inspired by the FITS linear forecast model; an effective lightweight approach for time series forecasting (Xu et al., 2023). FITS applies the Real Fourier Transform (RFT) to a context vector, removes high frequency components using a low-pass filter (LPF) and then takes the inverse RFT to map back into the time domain. This structure is similar to that of the ELF-Forecaster, however there are some important differences between our ELF-Forecaster and FITS, some of which we summarise below:

1. **Model Fitting:** While FITS is fit using gradient descent, ELF uses a closed-form solution to find the complex weight matrix.

2. **Model Output:** FITS outputs a forecast of the target and a reconstruction of the context vector, whereas ELF only generates a target.

3. **Loss Function:** FITS is trained to minimise both the reconstruction error and forecast error where we only consider forecast error.

4. **Target Compression:** FITS uses a low-pass filter to compress the context vector. ELF applies an LPF to the context and target.

5. **Online Learning:** Our method is designed for online learning whereas FITS considers exclusively the non-online setting.

**Relation To Standard Linear Regression**   When FITS does not use a LPF and the context length exceeds the prediction horizon length, it is known that it is equivalent to ordinary least-squares linear regression (Toner & Darlow, 2024). Similarly, in the case where our model applies no compression to the target or the context it will also be equivalent to ordinary least-squares linear regression.

### A.1.3. Fitting and Forecasting Algorithms

We present here the algorithms which form the ELF-Forecaster. Algorithm 1 describes how the ELF-Forecaster generates a forecast given an input context vector $x \in \mathbb{R}^L$. Algorithm 2 describes how the ELF-Forecaster is refit at update step $\tau$ (i.e. after $\tau M$ time steps).

---

**Algorithm 1** Predicting with ELF-Forecaster

---

**Input:** Context vector $\boldsymbol{x} = (x_1, x_2 \ldots, x_L)$, forecast horizon $H$, frequency retention proportion $\alpha \in [0, 1]$, complex weight matrix $W \in \mathbb{C}^{\tilde{L} \times \tilde{H}}$, where $\tilde{L} := \lfloor \alpha L \rfloor$ and $\tilde{H} := \lfloor \alpha H / 2 \rfloor$.

Compute mean of input: $\mu := \frac{1}{L} \sum_{i=0}^{L-1} x_i$
Normalise data: $\boldsymbol{x} \leftarrow \boldsymbol{x} - \mu$
Apply discrete Fourier transform (DFT): $\boldsymbol{x} \leftarrow \mathrm{DFT}(\boldsymbol{x})$
Discard high-frequency components by removing the central portion of $\boldsymbol{x}$:

$$\boldsymbol{x} \leftarrow (x_1, x_2, \ldots, \underbrace{x_{\frac{\alpha L}{2}}, \ldots, x_{L - \frac{\alpha L}{2}}}_{\text{Remove } L - \lfloor \alpha L \rfloor \text{ components}}, \ldots, x_L)$$

Apply linear map: $\hat{\boldsymbol{y}} \leftarrow \boldsymbol{x}^T W$
Pad forecast with zeros to length $H/2 + 1$:

$$\hat{\boldsymbol{y}} \leftarrow (\hat{y}_1, \ldots, \hat{y}_{\tilde{H}}, 0, \ldots, 0) \in \mathbb{C}^{H/2+1}$$

Apply inverse real Fourier transform (iRFT): $\hat{\boldsymbol{y}} \leftarrow \mathrm{iRFT}(\hat{\boldsymbol{y}})$
Restore mean: $\hat{\boldsymbol{y}} \leftarrow \hat{\boldsymbol{y}} + \mu$
**Return** Forecast vector $\hat{\boldsymbol{y}} \in \mathbb{R}^H$

---

---

**Algorithm 2** Fitting of ELF-Forecaster

---

**Input:** Update step $\tau$, refit interval $M = 200$; context length $L$; forecast horizon $H$; the last $M + L + H$ time steps of a time series $\{x_i\}_{i=(\tau-1)M-L-H}^{\tau M-1}$; frequency retention proportion $\alpha \in [0, 1]$; regularisation parameter $\lambda = 20$; number of datapoints observed so far NumSeen; boolean flag FIRSTFIT (TRUE if the model has not yet been fit)

Denote
$$\boldsymbol{x}_s \coloneqq (x_s, x_{s+1}, \ldots, x_{s+L-1}) \quad \boldsymbol{y}_s \coloneqq (x_{s+L}, x_{s+L+1}, \ldots, x_{s+L+H-1})$$

Gather the last $M$ context and target vectors $\mathcal{D} \coloneqq \{\boldsymbol{x}_s, \boldsymbol{y}_s\}_{s=\tau M-L-H-M}^{\tau M-L-H}$.

Form matrices $X \in \mathbb{R}^{M \times L}$ and $Y \in \mathbb{R}^{M \times H}$ from $\mathcal{D}$.

Normalise $X \leftarrow X - \mathrm{mean}(X, \mathrm{axis} = 1)$          // Normalise rows to have zero mean

**if** FIRSTFIT **then**

    $\sigma \leftarrow \mathrm{stdev}(X, \mathrm{dim} = 1)$          // Calculate initial standard deviation per feature

**else**

    $\sigma \leftarrow \mathrm{WELFORD}(X, \sigma)$          // Update standard deviation online

**end if**

$X \leftarrow X/\sigma$          // Scale data using standard deviation for numerical stability

*// Transform to frequency domain*

$\tilde{X} \leftarrow \mathrm{DFT}(X)$          // Apply row-wise discrete Fourier transform

$\tilde{Y} \leftarrow \mathrm{RFT}(Y)$          // Apply row-wise real Fourier transform

*// Reduce dimensionality of $\tilde{X}, \tilde{Y}$*

Remove the last $\frac{H}{2} - \lfloor \frac{\alpha H}{2} \rfloor$ columns of $\tilde{Y}$          // Low-pass on RFT output (ordered low-to-high)

Remove the *middle* $L - \lfloor \alpha L \rfloor$ columns of $\tilde{X}$          // Low-pass on DFT output (high freqs in middle)

**if** FIRSTFIT **then**

    $A^{-1} \leftarrow \left( \frac{\tilde{X}^* \tilde{X} + \lambda I}{M} \right)^{-1}$

    $B \leftarrow \frac{1}{M} \tilde{X}^* \tilde{Y}$

    FIRSTFIT $\leftarrow$ FALSE

**else**

    *// Apply complex-valued Woodbury update rule*

    $A^{-1} \leftarrow \mathrm{WOODBURYUPDATE} \left( \frac{A^{-1}}{\mathtt{NumSeen}}, \tilde{X} \right)$          // Update $A^{-1}$ efficiently; see Alg. 3

    $A^{-1} \leftarrow A^{-1} \cdot (\mathtt{NumSeen} + M)$          // Rescale updated $A^{-1}$

    $B \leftarrow \frac{1}{\mathtt{NumSeen}+M} \left( \mathtt{NumSeen} \cdot B + \tilde{X}^* \tilde{Y} \right)$          // Update $B$ via weighted average

**end if**

Update counter: $\mathtt{NumSeen} \leftarrow \mathtt{NumSeen} + M$

Set the complex weight matrix $W \leftarrow A^{-1}B$

---

---

**Algorithm 3** Complex-Valued Woodbury Update (Woodbury, 1950)

---

**Function:** WOODBURYUPDATE$(A^{-1}, X)$

**Input:** $A^{-1} \in \mathbb{C}^{d \times d}$ (inverse of a $d \times d$ matrix), $X \in \mathbb{C}^{M \times d}$

Compute the intermediate matrix:
$$B \leftarrow A^{-1}X^* \left( I + XA^{-1}X^* \right)^{-1} XA^{-1}$$

Update inverse:
$$A^{-1} \leftarrow A^{-1} - B$$

**Return** updated inverse matrix $A^{-1}$

---

## A.2. ELF-Weighter Algorithms

We present here the two algorithms which form the ELF-Weighter. Algorithm 4, shows how ELF-Weighter ELF the FM forecast by combining it with the ELF-Forecaster forecast using a weighted average. While Algorithm 5 shows how the ELF-Weighter uses exponential weighting to update the weights of the fast, slow and merge weighters which construct the weights $w_\tau$ used to combine the FM and ELF-Forecaster forecasts.

---

**Algorithm 4** ELF-Weighter Combined Forecasts at Update Step $\tau$

---

**Input:** $\hat{\boldsymbol{y}}_{t,FM}$ and $\hat{\boldsymbol{y}}_{t,EF}$, which are the forecasts of the FM and ELF-Forecaster for time $t$, respectively, and let the last update step be update step $\tau$

Compute weight:
$$w_\tau = \beta_\tau^{merge} w_\tau^{fast} + (1 - \beta_\tau^{merge}) w_\tau^{slow}$$

Compute and return ELF's forecasts:
**Return** $w_\tau \hat{\boldsymbol{y}}_{t,FM} + (1 - w_\tau)\hat{\boldsymbol{y}}_{t,EF}$

---

**Algorithm 5** ELF-Weighter Update at Update Step $\tau$

---

**Input:** $\widehat{Y}_{\tau,FM}$ and $\widehat{Y}_{\tau,EF}$; the $M \times H$ matrices containing as rows the $M$ rolling forecasts of the FM and ELF-Forecaster for update step $\tau$, respectively; and, $X_\tau$, $Y_\tau$ the $M \times L$ and $M \times H$ matrices containing the true values for the contexts and forecasts for update step $\tau$, respectively.

Compute losses for update step $\tau$ (we use average MASE over the last $M$ time steps for the loss function):
$$\text{Loss}_{\tau,1} = \text{Compute\_Average\_MASE}(\hat{Y}_{\tau,FM}, Y_\tau, X_\tau)$$
$$\text{Loss}_{\tau,2} = \text{Compute\_Average\_MASE}(\hat{Y}_{\tau,EF}, Y_\tau, X_\tau)$$
$$\text{Loss}_{\tau,s} = \text{Compute\_Average\_MASE}(w_{\tau-1}^{slow}\hat{Y}_{FM} + (1 - w_{\tau-1}^{slow})\hat{Y}_{\tau,EF}, Y_\tau, X_\tau)$$
$$\text{Loss}_{\tau,f} = \text{Compute\_Average\_MASE}(w_{\tau-1}^{fast}\hat{Y}_{FM} + (1 - w_{\tau-1}^{fast})\hat{Y}_{\tau,EF}, Y_\tau, X_\tau)$$

Update slow weighter:
$$w_\tau^{slow} = \frac{w_{\tau-1}^{slow} e^{-\eta \text{Loss}_{\tau,1}}}{w_{\tau-1}^{slow} e^{-\eta \text{Loss}_{\tau,1}} + (1 - w_{\tau-1}^{slow}) e^{-\eta \text{Loss}_{\tau,2}}}$$

Update fast weighter:
$$w_\tau^{fast} = \frac{e^{-\eta \sum_{\tau'=\tau-B}^{\tau} \text{Loss}_{\tau',1}}}{\sum_{j=1}^{2} e^{-\eta \sum_{\tau'=\tau-B}^{\tau} \text{Loss}_{\tau',j}}}$$

Update merge weighter:
$$\beta_\tau^{merge} = \frac{\beta_{\tau-1}^{merge} e^{-\eta \text{Loss}_{\tau,f}}}{\beta_{\tau-1}^{merge} e^{-\eta \text{Loss}_{\tau,f}} + (1 - \beta_{\tau-1}^{merge}) e^{-\eta \text{Loss}_{\tau,s}}}$$

---

# B. Hyperparameters and Additional Experimental Details

There are a few additional details to mention about our experiments. First, there are the hyperparameter values used for ELF. These are chosen *a priori* and are the same across all of our experiments. This is due to the fact in the rolling window setting it is not possible to fix hyperparameters before evaluating/deploying the forecaster. For the ELF-Forecaster the parameter values are $\lambda = 20$ and $\alpha = 0.9$. For the ELF-Weighter the hyperparameters are $\eta = 0.5$ for all weighters and $B = 5$ update steps. Second, at the start of deployment the ELF-Forecaster has insufficient data to give accurate forecasts therefore we have a warm-up period of 5 update steps where it is not used in the combined forecast. Last, in Table 3 we present the seasonalities used to calculate the MASE and RMSSE scores for each dataset along with some general dataset statistics—number of channels and time steps.

*Table 3.* **Dataset statistics and the seasonalities used for each dataset in the computation of MASE**

| Dataset | Seasonality | #Channels | #Time Steps |
|---|---|---|---|
| ETTh1 | 24 | 7 | 17420 |
| ETTh2 | 24 | 7 | 17420 |
| ETTm1 | 96 | 7 | 69680 |
| ETTm2 | 96 | 7 | 69680 |
| US Weather | 24 | 12 | 35064 |
| Weather | 144 | 21 | 52696 |
| Solar | 24 | 88 | 12840 |
| ECL | 24 | 321 | 26304 |
| Traffic | 24 | 862 | 17544 |
| Wind-PerSec | 60 | 1 | 86400 |
| Solar-PerSec | 60 | 1 | 86400 |
| Cloud-PerSec | 240 | 11 | 86400 |

**Details of FMs** For each of the FMs looked at, we have aimed to use the same configurations as in the original works. However, there are some necessary changes we needed to make. First, both TTM and TimesFM are trained using a context length of 512 and so in our experiments we remove the first 8 values of each 520-long context before giving it to TTM or TimesFM. Also, the currently released models for TTM only predict to a maximum horizon length of 96. Hence to forecast with TTM using a horizon length of 336 we auto-regressively feed-in the constructed forecast back into TTM to generate longer forecasts. This is the same technique as done in the paper proposing TTM (Ekambaram et al., 2024).

**Details of Online Adaption Methods: TAFAS, OneNet and FSNet** For the online adaptation methods we look at, as for the FMs, we have aimed to keep their setup the same as in the original works. Specifically, this means that for each method we have a training step, unlike ELF, which happens after seeing 2000 time steps. Before this training step each method is uninitialised, so we use the naive seasonal forecaster to give forecasts in this region. Then at the training step we do batch training on the first 2000 time steps of the dataset to initialise each method. After this the methods are trained online like ELF, where for TAFAS we set $M = 200$ to make it as comparable to ELF as possible. While, given how different OneNet, OneNet-TTM and FSNet are to ELF we set $M = 1$ for these three methods to maximise their performance, which is the same setting as in their original works (Zhang et al., 2023; Pham et al., 2023). Additionally, as we are in the rolling window setting we cannot normalise the data ahead of time, which is done in the original works proposing TAFAS, OneNet and FSNet. Therefore, instead we use RevIn (Kim et al., 2021) with each method to standardise the data in our experiments. Finally, there are a few more specific details we need to mention for TAFAS. One being that we do not use *Prediction Adjustment*. This is because in our setting we assume the forecasts cannot be modified after they have been given, as in the real-world they are often used immediately for decision making. While the other is that for TTM we use the both gated calibration modules for all datasets bar Solar and Traffic. For the other FMs and for Solar and Traffic for TTM we only use the output gated calibration module due to GPU Memory constraints we had while performing the experiments.

*Table 4.* **MASE of time series foundation models with and without using *ELF* on per-second time series:** A lower MASE is better and we present results for each dataset over multiple forecast horizon lengths denoted as $H$ in the table. The results show that by using *ELF* we improve performance across all per-second datasets and forecast lengths tested.

| Dataset | $H$ | Time Series FMs | | | | | | | | | |
|---|---|---|---|---|---|---|---|---|---|---|---|
| | | **TTM** | $+ELF(\downarrow)$ | **TimesFM** | $+ELF(\downarrow)$ | **VisionTS** | $+ELF(\downarrow)$ | **Chronos** | $+ELF(\downarrow)$ | **Moirai** | $+ELF(\downarrow)$ |
| Wind-PerSec | 30 | 0.728 | -0.041 | 0.774 | -0.089 | 1.492 | -0.794 | 1.096 | -0.398 | 0.757 | -0.070 |
| | 96 | 1.770 | -0.036 | 1.851 | -0.111 | 2.299 | -0.552 | 2.136 | -0.386 | 1.821 | -0.082 |
| | 336 | 4.759 | -0.058 | 4.923 | -0.208 | 4.975 | -0.260 | 5.219 | -0.482 | 4.786 | -0.079 |
| Solar-PerSec | 30 | 0.286 | -0.050 | 0.246 | -0.016 | 0.738 | -0.499 | 0.275 | -0.041 | 0.346 | -0.109 |
| | 96 | 0.581 | -0.103 | 0.508 | -0.030 | 0.907 | -0.417 | 0.536 | -0.051 | 0.666 | -0.183 |
| | 336 | 1.543 | -0.165 | 1.528 | -0.165 | 1.647 | -0.236 | 1.429 | -0.045 | 1.684 | -0.229 |
| Cloud-PerSec | 30 | 0.940 | -0.128 | 0.879 | -0.079 | 0.980 | -0.161 | 0.920 | -0.110 | 0.947 | -0.132 |
| | 96 | 1.060 | -0.119 | 1.010 | -0.085 | 1.075 | -0.026 | 1.067 | -0.127 | 1.082 | -0.137 |
| | 336 | 1.285 | -0.126 | 1.234 | -0.094 | 1.240 | -0.082 | 1.334 | -0.172 | 1.379 | -0.212 |

## C. Additional Experimental Results

### C.1. Results on Per-Second Datasets

In the main text we look at standard times series datasets used frequently in the literature (Darlow et al., 2024), here we look additionally at datasets which have a per-second frequency. The reason we do this is to explore settings where speed of updating and computing forecast is required to be measured in seconds and so where there is the most need for the compute efficiency of ELF. We perform experiments with three per-second datasets: Wind-PerSec (Godahewa et al., 2021), Solar-PerSec (Godahewa et al., 2021) and Cloud-PerSec (Joosen et al., 2023). The rest of the experimental setup is the same as for the experiments in the main text. Results for these experiments are presented in Table 4 and show that for all the per-second datasets using ELF leads to better performance. Hence, on high-frequency datasets where computational efficiency is a major factor, we find that the lightweight ELF improves forecasts.

### C.2. Comparison of ELF to TAFAS

In the main text we compare ELF with TAFAS using TTM as the FM. Here, we also provide results comparing ELF with TAFAS using the rest of the FMs used in our experiments, except for Chronos. The reason we do not report results for Chronos is due to its large computational cost. The results are presented in Table 5 and show that, as with TTM, ELF performs better than TAFAS on FMs tested for all datasets look at. This demonstrates the benefit of using ELF over TAFAS for the online adaptation of FM forecasts. However, we note that given the computational efficiency of both methods, it maybe more beneficial to use both methods at the same time to improve FM performance, which we leave to future work to explore.

### C.3. Comparison of ELF to Continual Finetuning

An alternative method to ELF for using online feedback in the rolling window setting is finetuning the FM at each update step. As explained in Section 2 this has three main problems: **a)** there is no default model-agnostic way to repeatedly finetune time series FMs; **b)** continually finetuning FMs lead to problems found in continual learning like catastrophic forgetting which are hard to deal with (De Lange et al., 2021; Lee & Storkey, 2024a); and **c)** finetuning large FMs is computationally expensive and so cannot be done in many real world deployment settings (Ekambaram et al., 2024). These reasons are why we do not compare the efficient, model-agnostic and unforgetting ELF to finetuning in the main text. However, it is still useful to see how well ELF compares to finetuning approaches.

To look at the performance of finetuning in the rolling window setting we perform an experiment where we finetune TTM at each update step. We use TTM as it is the only time series FM which we know of that proposes an efficient finetuning scheme but we note that this scheme is specific to TTM and cannot be used for other FMs, unlike ELF (Ekambaram et al.,

*Table 5.* **MASE of time series foundation models using *ELF* or *TAFAS*:** A lower MASE is better and we present results for each dataset over multiple forecast horizon lengths denoted as $H$ in the table. The results show that *ELF* performs better than TAFAS across all datasets and forecast lengths tested.

| Dataset | $H$ | Time Series FMs | | | | | | | |
|---|---|---|---|---|---|---|---|---|---|
| | | TTM | | TimesFM | | VisionTS | | Moirai | |
| | | *+ELF* | *+TAFAS* | *+ELF* | *+TAFAS* | *+ELF* | *+TAFAS* | *+ELF* | *+TAFAS* |
| ETTh1 | 30 | **0.911** | 0.927 | **0.891** | 0.910 | **0.904** | 0.960 | **0.922** | 1.009 |
| | 96 | **1.067** | 1.080 | **1.058** | 1.090 | **1.055** | 1.078 | **1.082** | 1.166 |
| | 336 | **1.280** | 1.289 | **1.288** | 1.317 | **1.283** | 1.300 | **1.316** | 1.418 |
| ETTh2 | 30 | **1.455** | 1.469 | **1.447** | 1.466 | **1.471** | 1.538 | **1.477** | 1.540 |
| | 96 | **2.770** | 2.788 | **2.759** | 2.789 | **2.759** | 2.786 | **2.801** | 2.872 |
| | 336 | **6.791** | 6.825 | **6.776** | 6.797 | **6.755** | 6.767 | **6.808** | 6.924 |
| ETTm1 | 30 | **0.753** | 0.781 | **0.754** | 0.828 | **0.768** | 1.008 | **0.767** | 1.072 |
| | 96 | **0.919** | 0.957 | **0.920** | 1.000 | **0.919** | 1.021 | **0.929** | 1.225 |
| | 336 | **1.143** | 1.193 | **1.145** | 1.230 | **1.139** | 1.199 | **1.152** | 1.459 |
| ETTm2 | 30 | **0.763** | 0.784 | **0.757** | 0.812 | **0.789** | 1.031 | **0.782** | 0.946 |
| | 96 | **0.953** | 0.979 | **0.951** | 1.015 | **0.964** | 1.065 | **0.971** | 1.151 |
| | 336 | **1.279** | 1.306 | **1.301** | 1.348 | **1.278** | 1.335 | **1.303** | 1.521 |
| US Weather | 30 | **0.857** | 0.893 | **0.827** | 0.857 | **0.855** | 1.008 | **0.828** | 0.896 |
| | 96 | **1.083** | 1.107 | **1.061** | 1.115 | **1.070** | 1.139 | **1.055** | 1.144 |
| | 336 | **1.252** | 1.269 | **1.241** | 1.303 | **1.238** | 1.274 | **1.224** | 1.336 |
| Weather | 30 | **0.855** | 0.880 | **0.777** | 0.795 | **0.893** | 1.322 | **0.900** | 1.138 |
| | 96 | **1.162** | 1.197 | **1.049** | 1.051 | **1.183** | 1.414 | **1.217** | 1.652 |
| | 336 | **1.536** | 1.567 | **1.493** | 1.613 | **1.550** | 1.643 | **1.579** | 2.021 |
| Solar | 30 | **1.060** | 1.091 | **1.038** | 1.096 | **0.984** | 1.004 | **1.091** | 1.222 |
| | 96 | **1.098** | 1.129 | **1.103** | 1.200 | **1.058** | 1.079 | **1.144** | 1.308 |
| | 336 | **1.134** | 1.166 | **1.149** | 1.248 | **1.149** | 1.236 | **1.173** | 1.292 |
| ELC | 30 | **0.917** | 1.000 | — | — | **0.875** | 0.977 | **0.937** | 1.223 |
| | 96 | **1.012** | 1.100 | — | — | **0.987** | 1.077 | **1.028** | 1.301 |
| | 336 | **1.193** | 1.267 | — | — | **1.191** | 1.298 | **1.212** | 1.474 |
| Traffic | 30 | **0.820** | 0.887 | — | — | **0.822** | 0.962 | **0.741** | 0.770 |
| | 96 | **0.843** | 0.921 | — | — | **0.832** | 0.943 | **0.735** | 0.752 |
| | 336 | **0.881** | 0.965 | — | — | **0.885** | 1.011 | **0.782** | 0.801 |

2024). To address the continual learning problems encountered by incrementally finetuning TTM at each update step, we finetune using experience replay, a well known and well performing continual learning method (Ostapenko et al., 2022; Chaudhry et al., 2019; Wang et al., 2024). More specifically, we set $M = 200$ as in our main experiments, maintain a memory buffer of 400 previously-seen instances—uniformly sampled from seen instances—and allow the method to see the last 400 time series instances. Then we finetune using TTMs scheme where the data in the memory buffer and the first 200 of the last 400 time series instances as training data and the last 200 being used as validation data for early stopping. We run the experiment on 2 Intel(R) Xeon(R) Platinum 8168 CPUs, to model a realistic deployment scenario, and measure both the MASE forecasting performance and the mean time taken to perform an update step for finetuning TTM and for ELF. The results of the experiments for the ETT datasets are presented in Table 6. The table shows that using ELF improves both forecasting accuracy and computational efficiency. For example, ELF has a 5.89% better forecasting performance and is 2506x faster. This means that while finetuning may not be able to be used in a real-world deployment setting due to computational expense (Joosen et al., 2023; Diao et al., 2024), ELF very likely can be as it incurs only a very small computational overhead.

*Table 6.* **Computational and forecasting performance of using ELF with TTM compared to incrementally finetuning TTM (TTM-Finetune):** In the table we present the average results for all the ETT datasets over the forecast horizons of $\{30, 96, 336\}$. The results show that using ELF improves both the forecasting performance and computational efficiency when compared to finetuning.

| Method | Seconds-Per-Update (CPU) | MASE |
|---|---|---|
| **TTM-Finetune** | 911.52 | 1.746 |
| **TTM+*ELF*** | 0.38 | 1.673 |
| Avg. *ELF* improvement | **2506x** ($\uparrow$) | **5.89%** ($\uparrow$) |

*Table 7.* **MASE results of finetuning TTM once on the ETT datasets (*TTM-Single-Finetune*):** For each dataset and forecast horizon, we finetune TTM on the first 2000 time steps and the use it to forecast the rest of the time series. We compare this method to (zero-shot) TTM and when using TTM with ELF (TTM+*ELF*). Furthermore, we bold methods which perform the best for each dataset and forecast horizon combination. The table shows that finetuning TTM once damages its performance, as this performs worse than TTM without finetuning.

| Dataset | $H$ | TTM | TTM-Single-Finetune | TTM+*ELF* |
|---|---|---|---|---|
| ETTh1 | 30 | 0.930 | 0.965 | **0.911** |
|  | 96 | 1.081 | 1.172 | **1.067** |
|  | 336 | 1.286 | 1.537 | **1.280** |
| ETTh2 | 30 | 1.472 | 1.579 | **1.455** |
|  | 96 | 2.786 | 2.927 | **2.770** |
|  | 336 | 6.802 | 7.044 | **6.791** |
| ETTm1 | 30 | 0.802 | 0.856 | **0.753** |
|  | 96 | 0.973 | 1.055 | **0.919** |
|  | 336 | 1.205 | 1.346 | **1.143** |
| ETTm2 | 30 | 0.799 | 0.885 | **0.763** |
|  | 96 | 0.991 | 1.116 | **0.953** |
|  | 336 | 1.320 | 1.564 | **1.279** |

### C.4. Comparison of ELF to a Single Finetuning Step

In the previous section we explored how well ELF compared to finetuning TTM at every update step. In this section we explore how well it compares to finetuning TTM once, a commonly explored setup in previous work (Ekambaram et al., 2024; Ansari et al., 2024). More specifically, we use TTM zero-shot for 2000 time steps and then use the data seen to finetune TTM. For the ETTh datasets this corresponds to finetuning on $\approx 10\%$ of the data. We use the same method as in Appendix C.3 to perform the finetuning with the newest 400 data points used for validation and the rest for training. After this we use the finetuned TTM to forecast the rest of the time series. The results of these experiments for the ETT datasets are presented in Table 7. The table shows that by only finetuning once we perform worse than using TTM zero-shot and additionally worse than using TTM with ELF. This indicates that while a single finetuning step might help to improve forecasts in the short term, in the long term it can damage TTMs ability to generalise to the changing data distribution of the time series. Therefore, these results suggest the need for online updating to improve forecasts of FMs at deployment.

### C.5. Continually Finetuning TTM with ELF

Up to this point we have assumed that we keep the FM fixed while using ELF. This is mainly due to the fact, as discussed before, that it is computationally expensive to continually finetune an FM. Additionally, the continual finetuning of FMs is not straightforward, coming with numerous complications as demonstrated by work in continual learning (CL) (De Lange et al., 2021). However, it is still interesting to explore the performance of using ELF while continuing finetuning the FM. To do this we perform an experiment where we continually finetune TTM and use ELF (TTM-Finetune+*ELF*) on the ETT datasets. We use TTM in this experiment as it is the only FM, to our knowledge, with a efficient finetuning scheme. The

*Table 8.* **MASE results of finetuning TTM and using ELF (*TTM-Finetune+ELF*) on the ETT datasets :** We report results where we continually finetune TTM alongside using ELF to adapt its forecasts. We bold methods which perform the best for each dataset and forecast horizon combination. The table shows that finetuning TTM with ELF in general performs worse than keeping TTM frozen.

| Dataset | $H$ | **TTM+*ELF*** | **TTM-Finetune+*ELF*** |
|---|---|---|---|
| ETTh1 | 30 | **0.911** | 0.940 |
| | 96 | **1.067** | 1.081 |
| | 336 | **1.280** | 1.300 |
| ETTh2 | 30 | **1.455** | 1.595 |
| | 96 | **2.770** | 2.784 |
| | 336 | 6.791 | **6.784** |
| ETTm1 | 30 | **0.753** | 0.784 |
| | 96 | 0.919 | **0.918** |
| | 336 | 1.143 | **1.140** |
| ETTm2 | 30 | **0.763** | 0.799 |
| | 96 | **0.953** | 0.959 |
| | 336 | **1.279** | 1.282 |

results of this experiment are presented in Table 8 and show that keeping TTM fixed performs best in most cases. However, for a few dataset and forecast horizon combinations finetuning TTM alongside ELF helps, though is more computationally costly. Therefore, this experiments shows that keeping the FM fixed is a sensible design decision, performing in general better. But, there maybe cases where computational cost and continual learning are not as big issues where it is beneficial to finetune the FM as well.

## C.6. ELF Performance versus Update Frequency

In our experiments we refit the ELF-Forecaster every $M = 200$ time steps. This number is fixed across our experiments and has not been tuned. In this section we look at how performance is impacted by varying this update parameter. Figure 5 plots the MASE of our approach as we increase the update frequency of ELF. The results shown are for the ETTh2 (right) and ETTm1 (left) datasets for the TTM FM (the FM attaining the best results on this dataset). These graphs demonstrate how more regular updating generally improves performance of our approach. This supports our core hypothesis that for time series it is crucial to utilise the most up-to-date data.

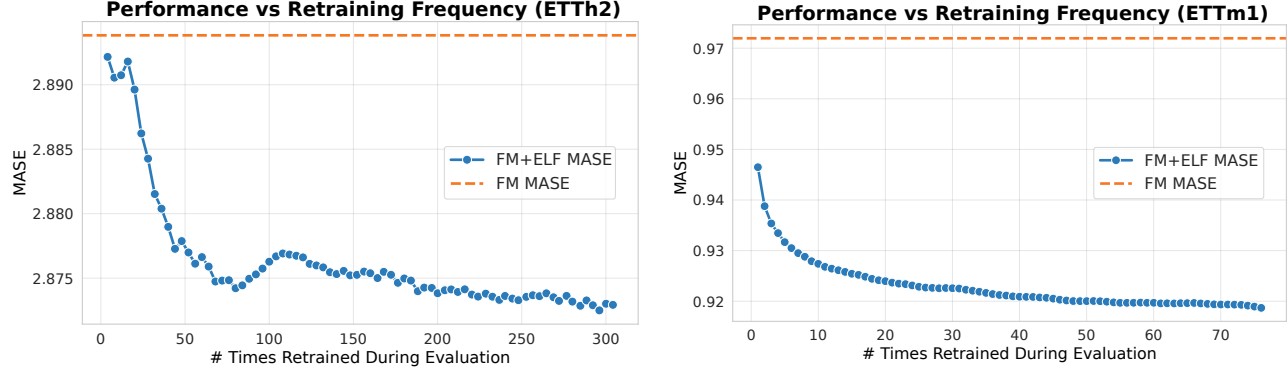

*Figure 5.* **ELF performance as a function of the update frequency used during online evaluation, indicating that more frequent updating boosts performance:** The left figure shows results for the ETTh2 dataset, while the right figure shows results for the ETTm1 dataset.

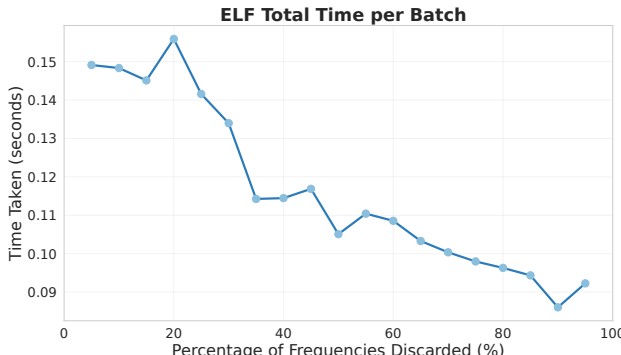

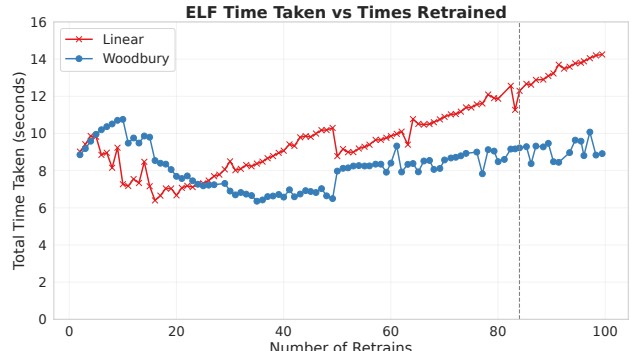

*Figure 6.* **The impact on time taken for fitting and inference as a function of percentage of frequencies discarded by ELF:** We measure the time ELF takes for fitting and inference on a batch size of 200 as the percentage of discarded frequency components increases. As more frequencies are discarded, the total time decreases. Fitting and inference are performed using two CPU cores. Due to the presence of noise, the curve is not perfectly monotonic, as might be expected.

*Figure 7.* **The impact of retraining frequency on ELF fitting+inference speed:** We plot the total time taken for fitting and inference in ELF as we vary the number of times the model is refitted during evaluation on the ETTh1 dataset with a horizon of 336. We compare the performance of the Woodbury update rule (blue) with the more basic OLS ('linear') approach. No frequency components are removed in this experiment. As retraining becomes more frequent, the benefit of using the Woodbury identity increases; when retraining occurs 84 times, the speed-up is approximately 25% over the naive 'linear' implementation. Minor fluctuations in the curve reflect the noise inherent in measuring computation speed.

## C.7. ELF-Forecaster Computational Efficiency Ablation

When training online in real-time is it vital that model fitting be fast and computationally low-cost. This necessity motivates our decision to use a linear model which can be fit in closed-form for the ELF-Forecaster. Additionally, as outlined in Section 4, we take advantage of two ways to speed up our method: **1)** we use the Woodbury matrix identity and **2)** we discard high frequency components when fitting the ELF-Forecaster. In this section we evaluate how these techniques impact the amount fitting time on a CPU. This ablation complements Section C.8 which explores how removing high frequencies impacts the performance of ELF, demonstrating that the decline in performance is slight. Thus together this section and Section C.8 validate our decision to remove high frequencies to improve speed.

### C.7.1. IMPACT OF % OF DISCARDED FREQUENCIES ON EXECUTION SPEED

As detailed in Section 4 the ELF includes a feature allowing the model to discard a percentage of the high frequencies of the context and targets when fitting in order to improve inference and fitting time. In this section we explore how varying the percentage of high frequencies which are discarded impacts the total execution time for fitting and inference. Specifically, we vary the proportion of frequencies discarded in steps of $5\%$ and record the time taken by ELF, for fitting and inference on a single data batch of size 200. For these experiments we do not use the Woodbury identity to investigate the role of frequencies on execution time in isolation. To ensure consistent resource allocation, we bound the execution to two CPU cores. This allowed us to isolate the computation to specific cores, mitigating potential variability caused by the operating system's task scheduler.

Figure 6 shows the results of these experiments, plotting fitting + inference time against percentage of discarded frequencies. The plot suggest that execution speed improves by discarding a higher proportion of frequency components. For example, discarding 40% of components results in a 25% speed-up compared to using 100% of the components.

### C.7.2. SPEED-UP FROM THE WOODBURY MATRIX IDENTITY

The ELF-Forecaster, detailed in Section 4, uses the Woodbury matrix identity. In this section we record how this design decision impacts the time taken for fitting and inference of ELF. We vary the number of times that we refit the ELF-Forecaster during evaluation on the ETTh1 dataset and record the total time taken to fit and predict using the ELF model. We repeat twice, once using the Woodbury identity and once not using the identity which we call *linear*. All fitting and prediction

occurs on 2 CPUs. For these experiments we do not throw away any high frequency components so that we can study the impact of the woodbury identity in isolation. We use a prediction horizon of 336. The results of these experiments are plotted in Figure 7, where we plot the number of retrains on the x-axis aginst total time in the y-axis. 'Woodbury' is plotted in blue and 'linear' in red.

By inspection of Figure 7 we see that, when one retrains rarely, it is preferable *not* to use the Woodbury update rule. This is because the Woodbury update rule is suited for low-rank updates and when refitting occurs infrequently, the rank of the update matches the rank of the matrix being updated. However, when retraining occurs regularly the Woodbury grants a speed-up over the naive implementation. Moreover, the gain increases as the regularity of retraining increases. In our main experiments we refit every 200 times steps. For the ETTh1 dataset this corresponds to refitting the ELF-Forecaster 84 times. At this retraining frequency the Woodbury matrix identity grants a roughly 25% speed-up. The graph features certain bumps which reflect the noise inherent in measuring execution time. Both graphs show a bump at 50 retrains. On the ETTh1 dataset 50 retrains corresponds to retraining every 336 time steps, this value is equal to the prediction horizon length. Consequently, as the number of retrains increases from 49 to 50 prediction length exceeds the batch size for the first time impacting the speed of the matrix computations and explaining the apparent discontinuity.

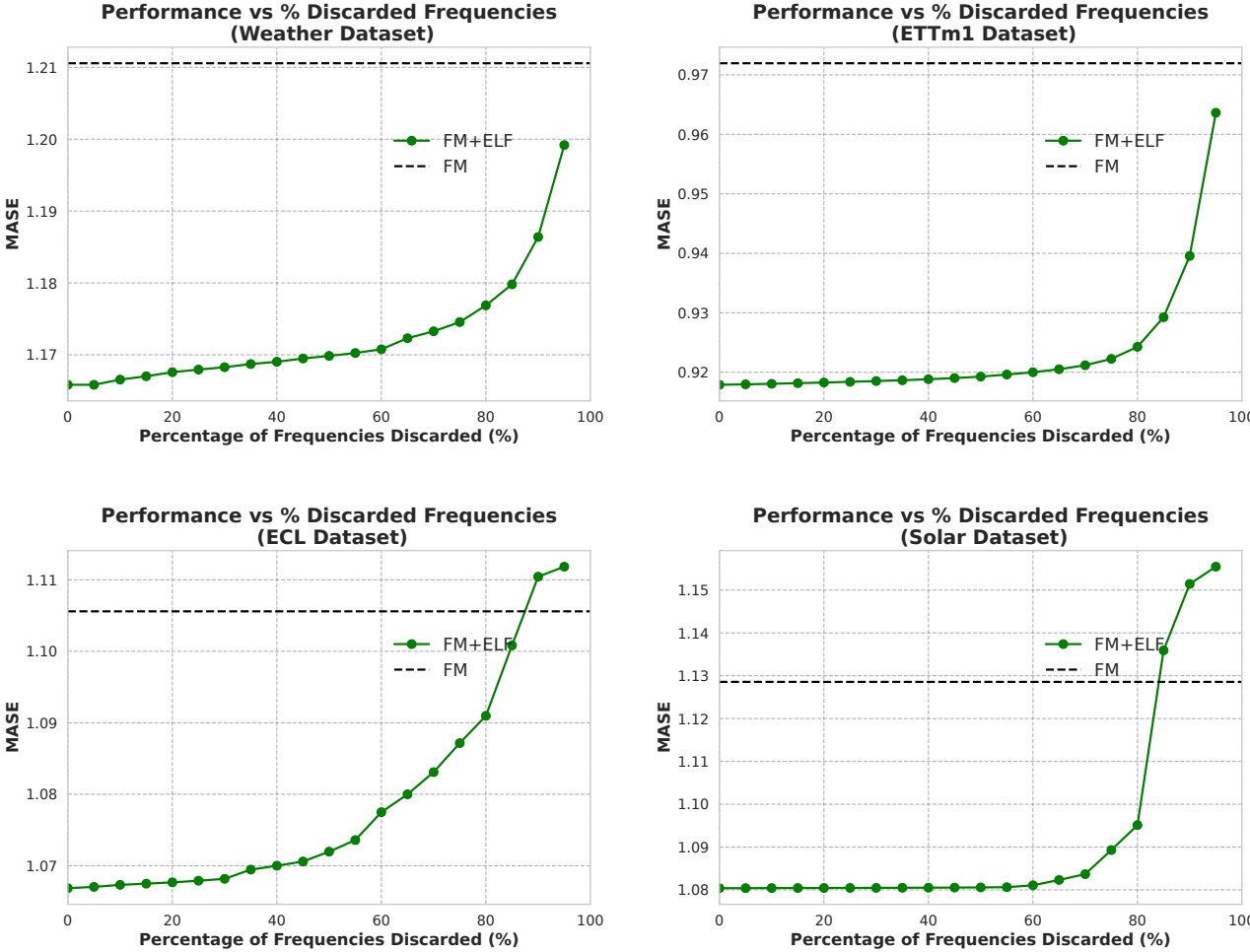

*Figure 8.* **ELF performance as a function of the percentage of high-frequency components discarded before model fitting:** We plot the MASE of FM+ELF as the percentage of high-frequency components removed by the ELF-Forecaster increases, for the Weather (top left plot), ETTm1 (top right), Electricity (bottom left), and Solar (bottom right) datasets. The foundation model (FM) used is TTM. The MASE of the FM (black dashed line) is shown for comparison. As more frequencies are discarded, performance begins to decline; however, initially, this decline is minor (e.g. for the ETTm1 dataset, removing 20% of frequency components results in only a roughly 0.1% increase in MASE).

### C.8. ELF Performance as Function of % High-Frequencies Discarded

The ELF-Forecaster removes high frequencies from the context and target to reduce dimensionality, thereby speeding-up model fitting and inference. In Figure 8 we look at how performance is impacted by this design choice. We plot the performance on the y-axis and the percentage of frequency components which are discarded on the x-axis. The plot shows the Weather (top left plot), ETTm1 (top right), Electricity (bottom left), and Solar (bottom right) datasets, each using a forecast horizon of 96 and a TTM base forecaster. The performance of the TTM FM without online adaption is given by the horizontal black dashed line. While removing high frequencies results in a drop in performance, removing only 10-20% of frequencies has a negligible impact on method performance. Taking into account the meaningful speed-up observed in the ablations in Section C.7 validates this decision to drop high frequency components.

## C.9. ELF-Forecaster Ablation

It is important to understand the benefit of using the ELF-Forecaster when compared to other potential options. To do this we perform an experiment where we replace the ELF-Forecaster with different types of forecasters, keeping the rest of ELF the same. The different types of forecasters we look at are: a linear model trained with online gradient descent (OGD), FITS, FSNet and a linear forecaster trained online to predict the residuals of the FM. The results of this experiment are presented in Table 9. For brevity, we report for each FM and forecast horizon combination the average MASE relative to the the MASE of the naive seasonal forecaster, averaging over the datasets used in the main experiments. The table shows that for all forecast horizons and FMs the ELF-Forecaster outperforms all of the other forecasters tested. This provides validation for the design decisions taken for the ELF-Forecaster.

*Table 9.* **Ablation of using different forecasters instead of the ELF-Forecaster:** We replace the ELF-Forecaster with: a linear model trained by online gradient descent (OGD), FITS, FSNet and a residual predictor approach. In each case we change nothing except for the forecaster; combining the FM and the forecaster using the ELF-Weighter. Results are given as MASE relative to the the MASE of the naive seasonal forecaster and we average results across all datasets. We also report the standard error as subtext. The results show that using the ELF-forecaster gives better performance than using any of the other alternatives tested across each FM and forecast horizon.

| | | Average Rel. MASE w.r.t Naive Seasonal Forecaster over all Datasets | | | | |
|---|---|---|---|---|---|---|
| FM | $H$ | **ELF** | **OGD-Forecaster** | **FITS-Forecaster** | **FSNet-Forecaster** | **Residual-Predictor** |
| TTM | 30 | $\mathbf{0.796}_{\pm 0.038}$ | $0.840_{\pm 0.039}$ | $0.842_{\pm 0.040}$ | $0.833_{\pm 0.039}$ | $0.827_{\pm 0.043}$ |
| | 96 | $\mathbf{0.831}_{\pm 0.033}$ | $0.872_{\pm 0.029}$ | $0.874_{\pm 0.030}$ | $0.869_{\pm 0.030}$ | $0.871_{\pm 0.033}$ |
| | 336 | $\mathbf{0.867}_{\pm 0.028}$ | $0.904_{\pm 0.022}$ | $0.908_{\pm 0.022}$ | $0.905_{\pm 0.023}$ | $0.926_{\pm 0.014}$ |
| TimesFM | 30 | $\mathbf{0.776}_{\pm 0.052}$ | $0.817_{\pm 0.053}$ | $0.818_{\pm 0.054}$ | $0.814_{\pm 0.052}$ | $0.816_{\pm 0.054}$ |
| | 96 | $\mathbf{0.853}_{\pm 0.029}$ | $0.910_{\pm 0.029}$ | $0.912_{\pm 0.030}$ | $0.926_{\pm 0.018}$ | $0.905_{\pm 0.033}$ |
| | 336 | $\mathbf{0.898}_{\pm 0.017}$ | $0.958_{\pm 0.015}$ | $0.964_{\pm 0.016}$ | $1.014_{\pm 0.047}$ | $0.956_{\pm 0.014}$ |
| VisionTS | 30 | $\mathbf{0.791}_{\pm 0.030}$ | $0.930_{\pm 0.009}$ | $0.935_{\pm 0.008}$ | $0.918_{\pm 0.016}$ | $0.893_{\pm 0.013}$ |
| | 96 | $\mathbf{0.825}_{\pm 0.033}$ | $0.898_{\pm 0.029}$ | $0.902_{\pm 0.029}$ | $0.897_{\pm 0.033}$ | $0.883_{\pm 0.028}$ |
| | 336 | $\mathbf{0.868}_{\pm 0.027}$ | $0.928_{\pm 0.022}$ | $0.934_{\pm 0.022}$ | $0.930_{\pm 0.024}$ | $0.924_{\pm 0.021}$ |
| Chronos | 30 | $\mathbf{0.753}_{\pm 0.037}$ | $0.820_{\pm 0.032}$ | $0.822_{\pm 0.032}$ | $0.817_{\pm 0.033}$ | $0.816_{\pm 0.032}$ |
| | 96 | $\mathbf{0.810}_{\pm 0.041}$ | $0.915_{\pm 0.056}$ | $0.917_{\pm 0.056}$ | $0.914_{\pm 0.057}$ | $0.892_{\pm 0.048}$ |
| | 336 | $\mathbf{0.856}_{\pm 0.032}$ | $0.960_{\pm 0.046}$ | $0.966_{\pm 0.046}$ | $0.968_{\pm 0.051}$ | $0.933_{\pm 0.034}$ |
| Moirai | 30 | $\mathbf{0.800}_{\pm 0.041}$ | $0.939_{\pm 0.039}$ | $0.941_{\pm 0.050}$ | $0.928_{\pm 0.049}$ | $0.920_{\pm 0.050}$ |
| | 96 | $\mathbf{0.836}_{\pm 0.043}$ | $0.980_{\pm 0.029}$ | $0.982_{\pm 0.062}$ | $0.976_{\pm 0.062}$ | $0.947_{\pm 0.030}$ |
| | 336 | $\mathbf{0.870}_{\pm 0.037}$ | $0.990_{\pm 0.022}$ | $0.997_{\pm 0.054}$ | $0.996_{\pm 0.055}$ | $1.001_{\pm 0.019}$ |

## C.10. ELF-Weighter Ablation

The ELF-Weighter consists of a combination two separate weighers—fast and slow—therefore it is useful to understand the respective contribution of these two parts. To perform this ablation we ran an experiment where we only used the slow weighter (*ELF-SlowWeighter*) or fast weighter (*ELF-FastWeighter*) on their own. The results of this experiment is recorded in Table 10, where we report for each FM and prediction horizon the average MASE across all datasets used in our main experiments (ETTs, US-Weather, Weather, Solar, ECL and Traffic). The table shows, by the bolded values, that using the full ELF-Weighter is better or equal to only using the fast or slow weighter. Additionally, we find that only in a few cases (33%) does the performance of the fast weighter or slow weighter match the performance of the combined weighter. We find that in these cases the combined weighter defaults to either fast weighter or slow weighter, perhaps due to not finding a benefit in terms of performance of mixing them. Hence, we have shown that making the ELF-weighter a combination of a fast and slow weighter generally improves performance and never makes it worse, while also having minimal additional compute overhead.

In Table 10 we also show the results of using the Hedge weighting algorithm (*ELF-HedgeWeighter*) or an unweighted mean to combine the forecasts of the FM and ELF-Forecaster (*ELF-Unweighted*). Hedge is similar to exponential weighting

*Table 10.* **Results of ablation where we compare using the full ELF-Weighter (*ELF*) with when using only its slow weighter component (*ELF-SlowWeighter*), fast weighter component (*ELF-FastWeighter*), using Hedge (*ELF-HedgeWeighter*) or using an unweighted mean (*ELF-Unweighted*):** We present for each FM and prediction horizon the average MASE ($\pm$ standard error) across all datasets used in our main experiments. We bold the lowest avgerage MASE weighter each FM and prediction horizon. The table shows that using the full ELF-Weighter is better than using either of its components independently and that using Hedge or an unweighted mean performs poorly.

| FM | $H$ | Average Rel. MASE w.r.t Naive Seasonal Forecaster over all Datasets | | | | |
| --- | --- | --- | --- | --- | --- | --- |
| | | **ELF** | **ELF-SlowWeighter** | **ELF-FastWeighter** | **ELF-HedgeWeighter** | **ELF-Unweighted** |
| TTM | 30 | $\mathbf{0.796}_{\pm 0.038}$ | $0.797_{\pm 0.038}$ | $0.798_{\pm 0.038}$ | $0.809_{\pm 0.040}$ | $0.804_{\pm 0.037}$ |
| | 96 | $\mathbf{0.831}_{\pm 0.033}$ | $0.832_{\pm 0.034}$ | $0.833_{\pm 0.032}$ | $0.843_{\pm 0.034}$ | $0.839_{\pm 0.031}$ |
| | 336 | $\mathbf{0.867}_{\pm 0.028}$ | $0.868_{\pm 0.028}$ | $0.869_{\pm 0.027}$ | $0.877_{\pm 0.028}$ | $0.874_{\pm 0.025}$ |
| TimesFM | 30 | $\mathbf{0.776}_{\pm 0.052}$ | $0.782_{\pm 0.050}$ | $\mathbf{0.776}_{\pm 0.052}$ | $0.799_{\pm 0.054}$ | $0.784_{\pm 0.048}$ |
| | 96 | $\mathbf{0.853}_{\pm 0.029}$ | $0.858_{\pm 0.029}$ | $\mathbf{0.853}_{\pm 0.029}$ | $0.872_{\pm 0.030}$ | $0.871_{\pm 0.021}$ |
| | 336 | $\mathbf{0.898}_{\pm 0.017}$ | $0.900_{\pm 0.017}$ | $\mathbf{0.898}_{\pm 0.017}$ | $0.913_{\pm 0.017}$ | $0.967_{\pm 0.056}$ |
| VisionTS | 30 | $\mathbf{0.791}_{\pm 0.030}$ | $0.793_{\pm 0.030}$ | $0.792_{\pm 0.030}$ | $0.811_{\pm 0.034}$ | $0.829_{\pm 0.021}$ |
| | 96 | $\mathbf{0.825}_{\pm 0.033}$ | $0.828_{\pm 0.033}$ | $0.826_{\pm 0.032}$ | $0.841_{\pm 0.032}$ | $0.842_{\pm 0.031}$ |
| | 336 | $\mathbf{0.868}_{\pm 0.027}$ | $0.870_{\pm 0.027}$ | $0.870_{\pm 0.027}$ | $0.883_{\pm 0.027}$ | $0.878_{\pm 0.025}$ |
| Chronos | 30 | $\mathbf{0.753}_{\pm 0.037}$ | $0.758_{\pm 0.037}$ | $0.755_{\pm 0.036}$ | $0.777_{\pm 0.039}$ | $0.774_{\pm 0.32}$ |
| | 96 | $\mathbf{0.810}_{\pm 0.041}$ | $0.814_{\pm 0.040}$ | $0.811_{\pm 0.041}$ | $0.830_{\pm 0.041}$ | $0.841_{\pm 0.042}$ |
| | 336 | $\mathbf{0.856}_{\pm 0.032}$ | $0.858_{\pm 0.032}$ | $\mathbf{0.856}_{\pm 0.032}$ | $0.874_{\pm 0.029}$ | $0.886_{\pm 0.036}$ |
| Moirai | 30 | $\mathbf{0.800}_{\pm 0.041}$ | $0.802_{\pm 0.041}$ | $0.802_{\pm 0.042}$ | $0.818_{\pm 0.044}$ | $0.830_{\pm 0.037}$ |
| | 96 | $\mathbf{0.836}_{\pm 0.043}$ | $0.837_{\pm 0.043}$ | $0.838_{\pm 0.043}$ | $0.849_{\pm 0.043}$ | $0.872_{\pm 0.043}$ |
| | 336 | $\mathbf{0.870}_{\pm 0.037}$ | $\mathbf{0.870}_{\pm 0.036}$ | $0.872_{\pm 0.037}$ | $0.881_{\pm 0.036}$ | $0.901_{\pm 0.038}$ |

where the only difference is that at each update step the FM or ELF-Forecaster is chosen to solely give forecasts until the next update step by sampling from the distribution defined by the weights (Cesa-Bianchi & Lugosi, 2006). While, by an unweighted mean we mean that we set $w_\tau = 0.5$ for all update steps $\tau$. The results in Table 10 show that using either Hedge or an unweighted mean leads to poor performance and they are never better or equivalent to using the ELF-Weighter in our experiments. Hence, this provides evidence that using a weighted mean to combine forecasts is a good design decision.

### C.11. Analysis of the Performance of ELF-Forecaster Compared to FMs

To understand the impact of learning from the up-to-date feedback given in the rolling window setting, we present here the performance of using the ELF-Forecaster on it own. We compare this against using (zero-shot) TTM, the best performing FM in our experiments, and when using ELF with TTM (TTM+*ELF*). The results are displayed in Table 11, from which we can draw three main conclusions: **a)** for some datasets using TTM zero-shot performs better than learning from the given dataset online using ELF-Forecaster (e.g. ETTh1 and ETTh2); **b)** for other datasets by learning online on the given dataset ELF-Forecaster performs better than using TTM (e.g. ECL and Traffic); and **c)** using ELF-Forecaster to adapt the forecasts of TTM (TTM+*ELF*) always improves performance against using either separately. While using summary MASEs is informative, it is also useful to look at how the relative performance between the ELF-Forecaster and FMs changes over time. We can look at Figure 3 to do this, as it displays the weight $w_\tau$ used to weight between the ELF-Forecaster and the Chronos at each update step $\tau$ for ETTh1. The figure shows that for most channels that at the start Chronos is preferred and performs better than the ELF-Forecaster. But, as the ELF-Forecaster sees more data and therefore learns the specific time series characteristics it gradually performs better and therefore is weighted more heavily. This all shows that, as expected, using the up-to-date feedback in the rolling window setting (i.e. deployment stage), means we can learn a dataset specific forecaster (ELF-Forecaster) which steadily improves in performance over time to be comparable to the FM. Therefore, it can be used to adapt the FMs forecasts to be more dataset specific, to improve performance, which our results experimentally validate (e.g., see Appendix C.13).

*Table 11.* **MASE of the ELF-Forecaster, TTM and when adapting TTM with ELF (TTM+*ELF*):** The results show that ELF-Forecaster, which is trained online on each dataset, and TTM, performing zero-shot forecasting, perform similarly. TTM sometimes performs better than ELF-Forecaster and othertimes not. Additionally, TTM+*ELF* improves upon both TTM and using ELF-Forecaster on its own for all datasets and forecast horizon lengths.

| Dataset | $H$ | **ELF-Forecaster** | **TTM** | **TTM+*ELF*** |
|---------|-----|--------------------|---------|---------------|
| ETTh1 | 30 | 0.946 | 0.930 | **0.911** |
|  | 96 | 1.113 | 1.081 | **1.067** |
|  | 336 | 1.335 | 1.286 | **1.280** |
| ETTh2 | 30 | 1.503 | 1.472 | **1.455** |
|  | 96 | 2.819 | 2.786 | **2.770** |
|  | 336 | 6.822 | 6.802 | **6.791** |
| ETTm1 | 30 | 0.769 | 0.802 | **0.753** |
|  | 96 | 0.931 | 0.973 | **0.919** |
|  | 336 | 1.150 | 1.205 | **1.143** |
| ETTm2 | 30 | 0.793 | 0.799 | **0.763** |
|  | 96 | 0.981 | 0.991 | **0.953** |
|  | 336 | 1.300 | 1.320 | **1.279** |
| US Weather | 30 | 0.877 | 0.893 | **0.857** |
|  | 96 | 1.096 | 1.123 | **1.083** |
|  | 336 | 1.262 | 1.296 | **1.252** |
| Weather | 30 | 0.907 | 0.887 | **0.855** |
|  | 96 | 1.205 | 1.205 | **1.162** |
|  | 336 | 1.568 | 1.576 | **1.536** |
| Solar | 30 | 1.084 | 1.091 | **1.060** |
|  | 96 | 1.124 | 1.129 | **1.098** |
|  | 336 | 1.172 | 1.166 | **1.134** |
| ECL | 30 | 0.930 | 1.003 | **0.917** |
|  | 96 | 1.021 | 1.106 | **1.012** |
|  | 336 | 1.205 | 1.279 | **1.193** |
| Traffic | 30 | 0.871 | 0.887 | **0.820** |
|  | 96 | 0.888 | 0.920 | **0.843** |
|  | 336 | 0.922 | 0.965 | **0.881** |

## C.12. RMSSE Results

*Table 12.* **RMSSE of time series foundation models with and without using ELF:** A lower RMSSE is better and we present results for each dataset over multiple forecast horizon lengths denoted as '*H*' in the table. The results show that by using *ELF* we improve RMSSE scores across all datasets and forecast lengths tested, as when evaluating with MASE.

| Dataset | H | TTM | +ELF(↓) | TimesFM | +ELF(↓) | VisionTS | +ELF(↓) | Chronos | +ELF(↓) | Moirai | +ELF(↓) |
|---|---|---|---|---|---|---|---|---|---|---|---|
| | | | | | Time Series FMs | | | | | | |
| ETTh1 | 30 | 0.806 | -0.016 | 0.805 | -0.023 | 0.864 | -0.068 | 0.840 | -0.052 | 0.877 | -0.076 |
| | 96 | 0.954 | -0.012 | 0.994 | -0.049 | 0.975 | -0.031 | 1.017 | -0.068 | 1.032 | -0.074 |
| | 336 | 1.149 | -0.005 | 1.215 | -0.058 | 1.176 | -0.024 | 1.231 | -0.072 | 1.259 | -0.085 |
| ETTh2 | 30 | 0.838 | -0.015 | 0.842 | -0.021 | 0.911 | -0.072 | 0.868 | -0.034 | 0.896 | -0.056 |
| | 96 | 1.149 | -0.012 | 1.178 | -0.037 | 1.166 | -0.029 | 1.209 | -0.054 | 1.219 | -0.060 |
| | 336 | 1.807 | -0.010 | 1.832 | -0.036 | 1.802 | -0.015 | 1.897 | -0.086 | 1.900 | -0.084 |
| ETTm1 | 30 | 0.680 | -0.039 | 0.710 | -0.067 | 0.860 | -0.208 | 0.763 | -0.122 | 0.905 | -0.253 |
| | 96 | 0.865 | -0.046 | 0.910 | -0.088 | 0.963 | -0.139 | 1.040 | -0.216 | 1.093 | -0.265 |
| | 336 | 1.077 | -0.052 | 1.135 | -0.106 | 1.108 | -0.080 | 1.321 | -0.287 | 1.316 | -0.282 |
| ETTm2 | 30 | 0.668 | -0.032 | 0.693 | -0.055 | 0.868 | -0.213 | 0.742 | -0.097 | 0.805 | -0.154 |
| | 96 | 0.847 | -0.034 | 0.892 | -0.075 | 0.955 | -0.129 | 0.981 | -0.156 | 1.001 | -0.173 |
| | 336 | 1.123 | -0.036 | 1.187 | -0.094 | 1.163 | -0.073 | 1.285 | -0.186 | 1.299 | -0.198 |
| US Weather | 30 | 0.744 | -0.030 | 0.746 | -0.038 | 0.875 | -0.149 | 0.803 | -0.088 | 0.779 | -0.067 |
| | 96 | 0.971 | -0.032 | 1.035 | -0.092 | 1.015 | -0.072 | 1.063 | -0.115 | 1.030 | -0.087 |
| | 336 | 1.148 | -0.033 | 1.230 | -0.108 | 1.159 | -0.043 | 1.278 | -0.149 | 1.232 | -0.108 |
| Weather | 30 | 0.580 | -0.028 | 0.521 | -0.017 | 0.941 | -0.359 | 0.724 | -0.167 | 0.753 | -0.184 |
| | 96 | 0.987 | -0.035 | 0.888 | -0.034 | 1.203 | -0.223 | 1.363 | -0.402 | 1.337 | -0.365 |
| | 336 | 1.643 | -0.031 | 1.767 | -0.214 | 1.761 | -0.131 | 2.061 | -0.439 | 2.001 | -0.369 |
| Solar | 30 | 0.790 | -0.005 | 0.831 | -0.043 | 0.836 | -0.045 | 0.846 | -0.065 | 0.922 | -0.104 |
| | 96 | 0.866 | -0.006 | 0.957 | -0.079 | 0.919 | -0.045 | 0.997 | -0.106 | 1.031 | -0.131 |
| | 336 | 0.903 | -0.005 | 1.000 | -0.077 | 0.973 | -0.057 | 1.033 | -0.091 | 1.046 | -0.109 |
| ECL | 30 | 0.846 | -0.066 | — | — | 0.863 | -0.102 | 0.781 | -0.040 | 1.049 | -0.250 |
| | 96 | 0.956 | -0.070 | — | — | 0.964 | -0.092 | 0.929 | -0.063 | 1.135 | -0.235 |
| | 336 | 1.115 | -0.062 | — | — | 1.191 | -0.138 | 1.130 | -0.085 | 1.285 | -0.214 |
| Traffic | 30 | 0.709 | -0.051 | — | — | 0.793 | -0.128 | 0.587 | -0.027 | 0.630 | -0.026 |
| | 96 | 0.783 | -0.057 | — | — | 0.811 | -0.090 | 0.723 | -0.052 | 0.678 | -0.020 |
| | 336 | 0.843 | -0.057 | — | — | 0.876 | -0.090 | 0.895 | -0.115 | 0.754 | -0.021 |

## C.13. The ELF-Forecaster Learns Dataset-Specific Features

While FMs are designed to produce good forecasts out-of-the-box, avoiding a dataset-specific training routine can mean that such models are not optimally tuned to the given data distribution. In contrast, the ELF-Forecaster is fit to data drawn from the specific time series. This allows it to pick up on dataset-specific features than the FM may miss. This way combining the forecasts of the FM with those of the ELF-Forecaster can boost performance. Two concrete examples of this are given in this section.

Figure 9 shows forecasts on the Traffic dataset (channel 620) made by TTM (left, orange) and ELF-Forecaster (right, blue). We see that while the forecasts of TTM are good, capturing daily periodicity, it does not model the decrease in traffic which occurs during the weekend (regions shaded in blue). By contrast, the ELF-Forecaster is able to identify and predict this decrease in traffic.

Figure 10 shows forecasts on two cloud time series (Joosen et al., 2024) made by TTM (orange) and TTM+ELF (blue)

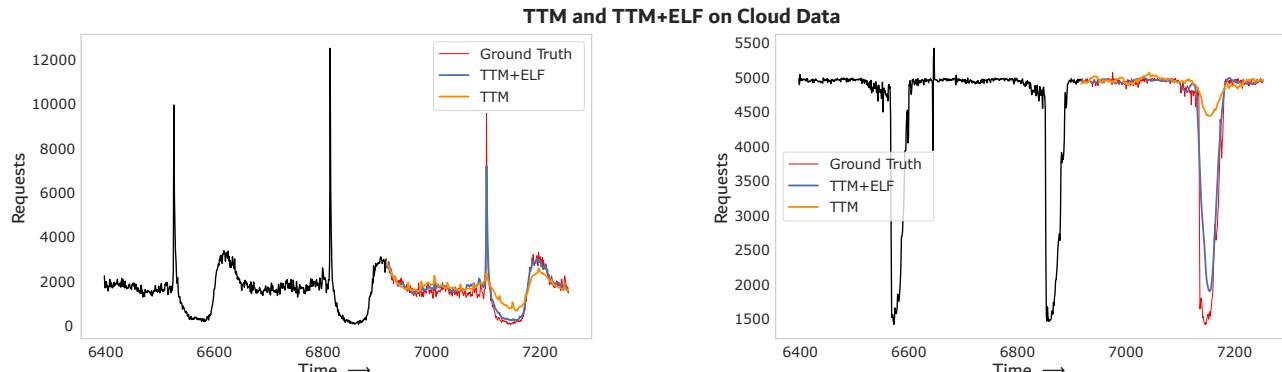

*Figure 9.* **TTM (left) and ELF-Forecaster (right) forecasts on the Traffic dataset:** The TTM forecast (right figure, orange) is good, picking up on the daily periodicity in traffic levels. However, TTM fails to model the weekly periodicity; specifically the decline in traffic occurring at the weekend (see the blue shaded regions). Conversely, the ELF-Forecaster, which is fit to the dataset, predicts this decline in weekend traffic numbers.

*Figure 10.* **TTM and TTM+ELF forecasts on two cloud time series:** In both figures, the TTM forecast (orange) is accurate but omits the large spikes which occurs around time step 7100 on the left figure and around step 7170 on the right figure. This is despite the regularity of these spikes making them seemingly easy to predict. In contrast, ELF predicts these spikes well so that TTM+ELF (blue) generates a superior forecast to TTM by itself.

compared against the ground truth (red). We see that while the forecasts of TTM are fairly accurate it does not anticipate spikes which occur periodically in either of the datasets. However, the ELF-Forecaster, which is fit on drawn from these dataset, is able to insert the missing spikes boosting overall performance.

## C.14. A Note on the Performance Ordering of FMs When Using or Not Using ELF

It is interesting to see how the relative performance of FM change when using ELF to adjust their forecasts online. The results in Table 1 show that in our experiments TTM is generally the best performing FM without ELF. But, when using ELF the best performing FM is less clear: the performance of each of the FMs becomes more similar and the best model varies across dataset and prediction length. This suggests that in realistic settings where it is possible to use ELF to exploit online feedback to improve forecasts, the performance improvement in newer FMs is less stark than in the zero-shot setting. This adds qualifications to the suggested progress made in time series forecasting by successive generations of FMs.

