# OpenReview forum: "Lightweight Online Adaption for Time Series Foundation Model Forecasts"
_ICML.cc/2025/Conference — ICML 2025 poster_

### Official Review · Reviewer_wUFv · 2025-02-27

**Overall Recommendation:** 4

**Summary:**

This paper identifies that existing foundation models (FMs) fail to fully utilize the large amount of online feedback obtained during the deployment phase. This is due to the high computational cost associated with regular retraining or fine-tuning, which often leads to the neglect of this valuable feedback. To address this issue, the authors propose a lightweight online adaptation mechanism named AdapTS, which enhances the performance of time-series forecasting tasks during the deployment phase of FMs by dynamically adjusting the prediction results of the FMs.

AdapTS consists of two main components: AdapTS-Forecaster and AdapTS-Weighter. The AdapTS-Forecaster employs a linear prediction model, which is updated using the mean squared error (MSE) loss to avoid potential issues associated with gradient optimization. It also leverages Fourier transforms to remove high-frequency components and utilizes the Woodbury matrix identity to achieve efficient updates. The AdapTS-Weighter, on the other hand, combines fast and slow weight mechanisms to dynamically adjust the prediction weights of the FM and AdapTS-Forecaster, thereby adapting to changes in data distribution.

## update after rebuttal

I am leaning towards accepting that even though this paper uses techniques from other fields applied to the time series field, I think it is innovative. I also hope that the authors enrich the related work section.

**Claims And Evidence:**

### Claim 1: lightweight and efficient

Firstly, in the design of the AdapTS-Forecaster component, a linear forecasting model is selected and fitted using the mean squared error (MSE) loss during each update. This approach avoids the complexities associated with gradient optimization. Secondly, the use of Fourier transforms to remove high-frequency components and the application of the Woodbury matrix identity enable efficient updates.

Therefore, it can be concluded that the paper's component design is centered around the goals of lightweight and high-efficiency, with efforts made to optimize and improve in these aspects.

Moreover, the experimental results demonstrate that AdapTS has high computational efficiency. Each update requires only an additional 0.38 seconds, which is 2,506 times faster than the existing online fine-tuning method (Online Fine-tuning TTM). This makes AdapTS suitable for use in resource-constrained environments.

### Claim 2: Enhancement of Online Prediction Performance of Time-Series Foundation Models (FMs) Using AdapTS

The results in Table 1 demonstrate that AdapTS efficiently and effectively utilizes online feedback in the rolling window setting, thereby improving the prediction results of the foundation models. The experimental findings show that AdapTS significantly enhances the forecasting performance of the FMs across multiple standard time-series datasets used as baselines. In some cases (e.g., when AdapTS adjusts the forecastings of VisionTS), the average improvement exceeds 10%.

### Claim 3: Can the AdapTS-Weighter be used to combine the forecasting  of FMs and AdapTS-Forecaster?

The experimental results in Table 5 of Appendix C.6 show that using the full AdapTS Weighter consistently achieves better results compared to the AdapTS unweighted experimental settings.

However, except on the ECL and Traffic datasets, the full AdapTS Weighter yields results that are mostly comparable to those of the fast weight mechanism in most cases. Therefore, the effectiveness of the AdapTS-Weighter component in dynamically adjusting weights to combine the forecasting of AdapTS-Forecaster and the foundation model (FM) is somewhat lacking in persuasiveness.

**Essential References Not Discussed:**

N/A

**Experimental Designs Or Analyses:**

### Rationality

To demonstrate the performance improvement brought by the AdapTS-Weighter component, the paper designed corresponding ablation experiments. In Table 5 of Appendix C.6, the full AdapTS-Weighter setting is compared with the AdapTS Unweighted setting, and the full AdapTS-Weighter consistently achieves better results.

### Questions

- In Section 5.5 Ablations, the paper only presents the conclusions drawn from the ablation experiments, with all experimental results relegated to the appendix. The main text lacks a systematic explanation of the reasons behind these conclusions (even a brief analysis of the specific numerical results would be helpful). As a result, the conclusions presented in Section 5.5 Ablations lack concrete experimental data to support them and are somewhat less convincing.
- The paper attempts to validate the rationale behind the decision to remove high-frequency components to improve speed by showing the proportion of discarded frequency components and performance on the Weather and ETTm1 datasets through Figure 8, in conjunction with Figure 6. However, there are some issues: Figure 8 only displays results from the Weather and ETTm1 datasets, and experimental results from more datasets are not shown in this paper. Therefore, the design of this ablation experiment is somewhat lacking in persuasiveness.

**Methods And Evaluation Criteria:**

The AdapTS-Forecaster component selects a linear forecasting model and achieves efficient updates through the use of Fourier transforms and the Woodbury matrix identity. The AdapTS-Weighter, on the other hand, combines fast and slow weight mechanisms to adapt to changes in data distribution. The design of the components focuses on efficiency, lightweight implementation, and rapid adaptation to changes in data distribution, which is rational.

**Other Comments Or Suggestions:**

N/A

**Other Strengths And Weaknesses:**

## Strengths

- Appendix C.1 clearly compares AdapTS, proposed in this paper, with the existing time-series FM (TTM) that is the only one to propose an effective fine-tuning scheme, demonstrating the advantages of AdapTS in generalization, efficiency, and performance.

- Section 5.4 provides a detailed explanation and reasoning for the improvement of online forecasting performance of time-series FMs by AdapTS, in conjunction with Figure 3 and Figure 4.

## Weakness

From the working mechanism of AdapTS-Forecaster, it can be inferred that its training data consist of the online feedback dynamically generated during the deployment phase. These data reflect the characteristics and changes of the current time series, enabling AdapTS-Forecaster to learn the most up-to-date data distribution. However, the paper lacks clarity and completeness in its design of “training data.” In particular, there are some omissions in the description of data processing and allocation in the experimental settings.

For more information, please see the Question For Authors section.

**Questions For Authors:**

- Neither the description of the experimental settings in the main text nor the detailed experimental information in Appendix B specifies what kind of online feedback data AdapTS, proposed in this paper, is based on. Does it assume that the online feedback data is reliable? If there is a specific explanation regarding this in the paper, please provide the exact location.

- ==Compared to the Prompt fine-tuning techniques that have been widely applied in computer vision and natural language processing [1-5], I believe your approach also involves adding lighter, trainable components to the frozen backbone of the model to achieve more efficient fine-tuning and better performance. Given this, I wonder if it is necessary to analyze the differences between your work and the existing rapid fine-tuning techniques in computer vision and natural language processing in the related work section. I am particularly curious about the challenges that might arise when transferring these techniques from the domains of image and natural language processing to time-series analysis, and how you have addressed these challenges. Alternatively, are there already similar works in these two domains that you have adapted to time-series foundation models? Please forgive my concerns because this technology is indeed very mature in other fields.==
- ==In addition to models used for zero-shot forecasting such as Moriai and Chronos, can the fine-tuning techniques you proposed be applied to pre-trained models for processing the five major tasks of TSA such as SymTime [] and UniTS []? And use this lightweight and efficient fine-tuning method for other tasks such as classification, filling and anomaly detection.==

[1] Lester, Brian, Rami Al-Rfou, and Noah Constant. "The power of scale for parameter-efficient prompt tuning." *arXiv preprint arXiv:2104.08691* (2021).

[2] Jia, Menglin, et al. "Visual prompt tuning." *European conference on computer vision*. Cham: Springer Nature Switzerland, 2022.

[3] Han, Cheng, et al. "E^ 2vpt: An effective and efficient approach for visual prompt tuning." *arXiv preprint arXiv:2307.13770* (2023).

[4] Sohn, Kihyuk, et al. "Visual prompt tuning for generative transfer learning." *Proceedings of the IEEE/CVF Conference on Computer Vision and Pattern Recognition*. 2023.

[5] Yao, Hantao, Rui Zhang, and Changsheng Xu. "Visual-language prompt tuning with knowledge-guided context optimization." *Proceedings of the IEEE/CVF conference on computer vision and pattern recognition*. 2023.

[6] Wang, Wenxuan, et al. "Mitigating Data Scarcity in Time Series Analysis: A Foundation Model with Series-Symbol Data Generation." *arXiv preprint arXiv:2502.15466* (2025).

[7] Gao, Shanghua, et al. "UniTS: A unified multi-task time series model." *Advances in Neural Information Processing Systems* 37 (2025): 140589-140631.

**Relation To Broader Scientific Literature:**

The AdapTS method holds significant importance in the field of time-series forecasting and is closely related to the literature on continual learning, online learning, and efficient computation. First, AdapTS enhances the performance of foundation models through its online adaptation mechanism, thereby avoiding the catastrophic forgetting problem often encountered in continual learning (De Lange et al., 2021; Lee & Storkey, 2024). Second, its design draws on classic methods in online learning, such as exponential weighting and the Woodbury matrix identity (Cesa-Bianchi & Lugosi, 2006; Rakhlin & Kleiner, 2008), and systematically applies them to time-series forecasting for the first time, effectively addressing dynamic changes in data distributions. Additionally, the lightweight design of AdapTS meets the demands of efficient computation, resolving the computational bottleneck associated with online fine-tuning of large-scale models (Ekambaram et al., 2024). Experiments demonstrate that AdapTS has broad adaptability across different datasets and models, aligning with the goals of zero-shot time-series forecasting (Ansari et al., 2024; Woo et al., 2024). In summary, AdapTS not only proposes an innovative method but also offers new perspectives for research in related fields.

**Theoretical Claims:**

### Statement 1: Three Reasons for Choosing Linear Prediction Models

The paper provides three reasons for selecting linear forecasting models as the AdapTS-Forecaster: a) They perform well in time-series forecasting (Zeng et al., 2023); b) They can be efficiently updated online according to the requirements of our setting; c) Unlike neural networks that are updated online, linear models do not encounter catastrophic forgetting during online updates (De Lange et al., 2021).

**Correctness of the Justifications:**

- **a) Good Performance:** The paper cites the study by Zeng et al. (2023) to support the good performance of linear models in time-series forecasting.
- **b) Efficient Updates:** Linear models are updated online via closed-form solutions, avoiding the computational burden of gradient descent. Parameter updates can be directly completed through matrix operations, which confirms the validity of this statement.
- **c) Avoiding Catastrophic Forgetting:** The paper references De Lange et al. (2021), noting that linear models update based on the entire data history and do not forget old data. In contrast, online updates of neural networks may lead to forgetting of old data, resulting in catastrophic forgetting.

### Statement 2: Upper Bound on Cumulative Loss of Exponential Weighting Methods

Theorem 4.1 (Cesa-Bianchi & Lugosi, 2006; Rakhlin & Kleiner, 2008): For convex loss functions, the cumulative regret of a weighted average predictor is given by:

$$R_T=\sum_{\tau=1}^T\mathrm{Loss}_{\tau,\text{weighted}}-\min_{k\in\{1,\ldots,K\}}\sum_{\tau=1}^T\mathrm{Loss}_{\tau,k}$$

When the maximum loss is *L*max and the learning rate is *η*=*L*max1*T*8ln*K*, the cumulative regret satisfies:

$$R_T \leq L_{\max} \sqrt{T \ln K}$$

**Correctness of the Proof:**

The paper cites the classic results from Cesa-Bianchi & Lugosi (2006) and Rakhlin & Kleiner (2008) to prove the upper bound on cumulative loss for exponential weighting methods under convex losses. This statement is correct.

### Statement 3: Adaptability Issues of Exponential Weighting Methods

Although exponential weighting methods are effective, they struggle to adapt quickly to distribution shifts (Jadbabaie et al., 2015; Cesa-Bianchi et al., 2012; Zhao et al., 2020).

**Correctness of the Proof:**

The paper references studies by Jadbabaie et al. (2015), Cesa-Bianchi et al. (2012), and Zhao et al. (2020) to highlight the limitations of exponential weighting methods in adapting to distribution changes. This statement is correct.

---

> ### Author Rebuttal · Authors · 2025-03-31
>
> Dear reviewer, thank you for your detailed review and constructive comments. We are happy that you thoroughly assessed our claims and found that they held up. We provide answers to your question below.
>
> **1. Adding details to ablations section**
>
> Thank you for raising this issue. We have updated the paper to expand the ablation section, including adding experimental data presented in a table, which we hope will address your concerns.
>
> **2. Ablation shown in Figure 8.**
>
> We agree that our rationale behind the decision to remove high-frequency components would be further enhanced by including more datasets to Figure 8. We have updated Figure 8 to include the Traffic and Solar datasets, and we find that the same conclusions hold.
>
> **3. Definition of online feedback**
>
> The online feedback we discuss in the paper is given by the dynamically arriving data points from the time series. Each data point gives feedback on the accuracy of the previous forecasts which aimed to predict the value of that data point (and others). This feedback is then used to update the AdapTS-Weighter and AdapTS-Forecaster. Additionally, we do not assume the data points are corrupted in any way other than that modelled by the time series itself (i.e. they are reliable). We tried to explain this on the fourth line of the second paragraph of the introduction but understand from your comment that this need to be more clear, especially in the "Rolling Window Forecasting" subsection which formally describes the setting we look at. Therefore, we have updated the paper to fix this issue.
>
> **5. Prompt and rapid fine-tuning techniques**
>
> We agree with you that there have been many methods proposed in vision and NLP to perform rapid finetuning of FMs and will add discussion of this to our related work section. There is however one large difference between vision and NLP and time series, which is that in time series linear models are still competitive. We exploit this fact in our construction of AdapTS in that the AdapTS-Forecaster is a lightweight linear model and that we do not require the adding of parameters or the backpropagating through the FM which prompt tuning and typical adaptor based methods require. This makes AdapTS computationally efficient and agnostic to the FM used. Additionally, looking at work on rapid finetuning of FMs, AdapTS would roughly fall into the adaptor based fine-tuning paradigm. As pointed out by the other reviewers there has been a work on using adaptor based continual finetuning in time series, i.e. TAFAS. We have now compared to TAFAS, with the results shown in the table in our comment to reviewer 6Y6Z. Our findings are that AdapTS outperforms TAFAS in all cases looked at.
>
> **6. Use of AdapTS in other TSA tasks**
>
> Thank you for suggesting these potential avenues to increase the scope of this work. While we focus on time series forecasting in this work and AdapTS is built for forecasting, it certainly is interesting to think whether it could be extended to perform continual adaption for other TSA tasks. To construct such a method for one of these tasks you would need some lightweight and online updatable method to replace the AdapTS-Forecaster. For example, for imputation you can use FITS. Then for generative tasks you can still use the AdapTS-Weighter. However, for predictive tasks you would need to use a slight modification whereby, as in the Hedge algorithm, you select a class/action using the weights as the probabilities instead of doing a weighted average. We see this direction as future work and want to focus this paper on time series forecasting but, given your comment, have added mention that AdapTS could be extended to these other tasks into the conclusions section.
>
> ****
> We would like to thank you again for your thoughtful review and hope that we have satisfactorily answered your remaining questions about the work.

---

### Official Review · Reviewer_GmRz · 2025-03-11

**Overall Recommendation:** 3

**Summary:**

This paper introduces AdapTS, a lightweight mechanism designed to enhance the adaptability of Foundation Models (FMs) for time series forecasting by incorporating online feedback. Traditional FMs remain fixed after deployment due to the high computational cost of online updates, preventing them from adapting to changing data patterns. To address this limitation, AdapTS consists of two key components: AdapTS-Forecaster, which learns the current data distribution to capture recent trends, and AdapTS-Weighter, which dynamically combines forecasts from both the FM and the AdapTS-Forecaster. The paper evaluates AdapTS across multiple benchmark time series datasets and demonstrates that it consistently improves forecasting accuracy.

**Claims And Evidence:**

Overall, the claims made in the paper are clear, but some aspects require further justification. Specifically, the decision to keep the foundation model fixed despite distribution shifts needs more explanation. Additionally, comparisons with prior adaptation-based forecasting methods should be included to support claims of novelty and contribution.

**Essential References Not Discussed:**

The paper appropriately discusses related work, and I did not identify any missing essential references.

**Experimental Designs Or Analyses:**

I have reviewed the experimental design and analyses, and they appear to be sound. However, additional experiments comparing AdapTS with adaptation-based forecasting methods such as FSNet, OneNet, and TAFAS would help better position the contribution.

**Methods And Evaluation Criteria:**

The proposed method's contribution—being fast and lightweight—needs additional justification, particularly in relation to the chosen benchmark datasets. Since the datasets used have relatively low sampling frequencies, the necessity of a fast adaptation mechanism is unclear. Further experiments on higher-frequency datasets or with more computationally intensive forecasters would strengthen the evaluation.

**Other Comments Or Suggestions:**

Please see Questions For Authors.

**Other Strengths And Weaknesses:**

Strengths
- The paper is well-written and clearly structured, making it easy to follow.
- The proposed framework is general and can be applied to various time series foundation models.
- Experimental results show consistent performance improvements across multiple datasets.

Weaknesses
- Some aspects of the method, including the decision to keep the foundation model fixed, require further justification (see Questions for Authors).
- The novelty and contribution of the proposed method compared to existing adaptation-based time series forecasting approaches need to be better clarified.
- The choice of benchmark datasets may not fully align with the claimed advantages of the method, particularly regarding computational efficiency.

**Questions For Authors:**

1. When the data distribution shifts, the performance of the foundation model itself is expected to degrade. Why is the foundation model kept fixed rather than updated? Wouldn't a two-online forecaster approach, such as OneNet [1], be more appropriate in this framework? The justification for using the fixed foundation models in this setup needs to be clarified.

2. Compared to OneNet, which replaces one of its online forecasters with a foundation model, what are the unique contributions and novelty of AdapTS? Additionally, experimental comparisons with adaptation-based time series forecasting methods (e.g., OneNet, FSNet [2], TAFAS [3]) would help contextualize the proposed approach.

3. The paper emphasizes the computational efficiency of AdapTS-Forecaster, yet the datasets used for evaluation have sampling frequencies in the range of minutes or hours. In such cases, rapid adaptation may not be as critical. Have you tested AdapTS on datasets with higher sampling frequencies? Additionally, how does AdapTS perform when using a more computationally expensive but advanced forecaster?

4. Why does linearly combining the forecasts from the foundation model and AdapTS-Forecaster lead to improved forecasting performance? Further theoretical or empirical justification would strengthen this claim.

5. The supplementary material discusses the differences between AdapTS and FITS [4]. However, given the structural similarities, it would be helpful to include an experimental comparison where FITS is used instead of AdapTS-Forecaster.

References
[1] OneNet: Enhancing Time Series Forecasting Models under Concept Drift by Online Ensembling
[2] Learning Fast and Slow for Online Time Series Forecasting
[3] Battling the Non-stationarity in Time Series Forecasting via Test-time Adaptation
[4] FITS: Modeling Time Series with 10k Parameters

**Relation To Broader Scientific Literature:**

The paper contributes to time series foundation models by introducing an adaptation mechanism that leverages online feedback. The proposed approach builds on prior work by incorporating a lightweight linear forecaster and weighting mechanism to improve forecasting performance.

**Theoretical Claims:**

I have checked the correctness of the theoretical claims presented in the paper.

---

> ### Author Rebuttal · Authors · 2025-03-31
>
> Thank you for your helpful comments and questions! We provide answers to your questions below and hope we have satisfactorily answered them, in particular by performing several additional experiments.
>
> **1. Why fixed FMs?**
>
> The main reason we keep the FM fixed is due to the computational expense of updating it. This is demonstrated by the fact AdapTS is 2506x faster than online finetuning TTM; the fastest FM to finetune. Additionally, online updating of FMs and neural networks more generally is not straightforward, coming with numerous complications as demonstrated by work in continual learning (CL). Furthermore, current CL solutions for these problems like FSNet are often complex and architecture specific. In contrast, one of the goals of AdapTS is to be simple and FM agnostic. Your comment illustrates that this point needed to be clarified in the paper which has been edited accordingly.
>
> To justify using a frozen FM instead of learning it online, we run an experiment where we finetune TTM alongside AdapTS. The results of this experiment (TTM-Fine+AdapTS) are given in the table for question 2. and show that it performs generally worse than AdapTS with TTM frozen (results are given relative to TTM+AdapTS). This alongside the vast compute benefits demonstrate why keeping the FM fixed is beneficial.
>
> **2. Comparisons with OneNet, FSNet and TAFAS**
>
> Thank you for pointing out the missing comparisons to OneNet, FSNet and TAFAS, which we agree should be compared to AdapTS. We have run these experiments as well and we provide the results for these baseline methods on the ETT datasets in the table below. Additionally, we provide results for OneNet where one of its forecasters is replaced with TTM (OneNet-TTM). We find that in all cases TTM+AdapTS performs better. Importantly, we perform much better than OneNet-TTM, our closest comparator. Importantly, AdapTS is also more computationally efficient than the other methods. We hope now to have better contextualised the effectiveness of AdapTS relative to these more compute-intensive adaptation methods.
> ||||Relative MASE to TTM+AdapTS||||
> |-|-|-|-|-|-|-|
> |**Dataset**|$H$|**TTM+TAFAS**|**OneNet**|**FSNet**|**OneNet-TTM**|**TTM-Fine+AdapTS**|
> |ETTh1|30|1.043|1.405|1.555|1.406|1.032|
> ||96|1.038|1.281|1.355|1.277|1.013|
> ||336|1.028|1.275|1.333|1.284|1.016|
> |ETTh2|30|1.030|1.273|1.359|1.259|1.096|
> ||96|1.020|1.169|1.184|1.167|1.005|
> ||336|1.006|1.196|1.102|1.333|0.999|
> |ETTm1|30|1.033|1.702|2.176|1.905|1.041|
> ||96|1.051|1.295|1.338|1.285|0.999|
> ||336|1.049|1.603|1.709|1.576|0.997|
> |ETTm2|30|1.041|1.769|2.025|1.764|1.047|
> ||96|1.038|1.421|1.596|1.502|1.006|
> ||336|1.025|1.657|2.359|1.555|1.002|
>
> Regarding the novelty of AdapTS compared to OneNet, the main contribution of AdapTS over OneNet is its superior computationally efficiency stemming from our ability to avoid performing online gradient optimization. The table indicates that this approach also provides a performance gain.
>
> **3. Higher frequency datasets and AdapTS performance when using a more advanced forecaster (or FITS)**
>
> We have run experiments using TTM on 3 per-second datasets: SWind, SSolar[5] and SCloud[6]. The results are given in the table below. As in the rest of our experiments we find that using AdapTS improves performance of FM forecasts. This shows that on high frequency datasets where computationally-fast adaptation is necessary, AdapTS still improves performance.
>
> We also looked at using a more advanced forecaster, FSNet, and FITS instead of the AdapTS-forecaster. The results are provided in the table given in response to reviewer 26Ko. We find that using FSNet or FITS instead of the AdapTS-Forecaster results in reduced performance. *We hope the results for FITS answers your 5th question.* The reason we believe FSNet performs poorly compared to the AdapTS-Forecaster is that the online updating of complex models like FSNet is to a large degree an unsolved problem as shown by results in continual learning. In contrast, learning the AdapTS-forecaster online is relatively easy, resulting in better online performance.
> |**Dataset**|$H$|**TTM**|*+AdapTS*|
> |-|-|-|-|
> |SWind|30|0.728|-0.041|
> ||96|1.77|-0.036|
> ||336|4.759|-0.058|
> |SSolar|30|0.286|-0.050|
> ||96|0.581|-0.103|
> ||336|1.543|-0.165|
> |SCloud|30|0.94|-0.128|
> ||96|1.06|-0.119|
> ||336|1.285|-0.126|
>
> **4. Why does linearly combining AdapTS-Forecaster and FM forecasts improve performance?**
>
> AdapTS corresponds to ensembling an FM (trained across numerous diverse time series) and a lightweight online forecaster (fit on recent time series data). The difference in training data means the forecasters are less correlated which is known to provide a performance benefit. Also, the weighter is able to optimally tune the ensemble weight online based on past performance leading to better performance than the individual models.
>
> [5] Monash Time Series Forecasting Archive
>
> [6] How does it function? characterizing long-term trends in production serverless workloads

---

> > ### Comment · Reviewer_GmRz · 2025-04-09
> >
> > Thank you to the authors for the clear responses and additional experiments.
> >
> > The new results help clarify my earlier concerns, particularly regarding the use of a fixed FM and comparisons with related methods such as OneNet, FSNet, and TAFAS. I appreciate the effort to contextualize AdapTS more thoroughly.
> >
> > I believe the paper would be strengthened by incorporating these comparisons into the related works and experimental section, to better position the contributions within the existing literature.
> >
> > In light of the clarifications and additional results, I am adjusting my score accordingly.

---

> > > ### Author Response · Authors · 2025-04-09
> > >
> > > Dear Reviewer,
> > >
> > > Thank you for your response, we appreciate you adjusting your review in light of our clarifications and additional experiments. We would like to point out that some further results you requested from our ongoing experiments are given in our most recent comment to reviewer 6Y6Z.
> > >
> > > Thank you again for your thoughtful comments and feedback. We believe that this process has greatly strengthened our work!

---

### Official Review · Reviewer_6Y6Z · 2025-03-12

**Overall Recommendation:** 3

**Summary:**

This paper proposes AdapTS, a lightweight method for the online adaptation of time series foundation model forecasts. It consists of an AdapTS-Forecaster and an AdapTS-Weighter. Experiments clearly show that AdapTS can significantly improve prediction performance across multiple models and datasets.

**Claims And Evidence:**

The claims in the paper are strongly supported by experiments. The authors conducted experiments using several foundation models and standard time series datasets. The notable reduction in the MASE when using AdapTS clearly indicates that it can remarkably enhance the prediction ability of foundation models.

**Essential References Not Discussed:**

There are no obvious missing essential references.

**Experimental Designs Or Analyses:**

The experimental designs are very sound. The rolling window setting closely mimics real - world deployment. Ablation experiments greatly help analyze the role of each part of AdapTS, and comparisons with other methods (such as fine - tuning) also strongly prove the effectiveness of AdapTS. However, the lack of comparison with Test time adaptation methods is an obvious and significant shortcoming.

**Methods And Evaluation Criteria:**

The proposed methods are highly reasonable. The two - part structure effectively addresses the problem of TSFM inability to adapt online. Using standard datasets and the MASE metric is extremely appropriate for evaluating the prediction performance of different time series.

**Other Comments Or Suggestions:**

None

**Other Strengths And Weaknesses:**

- **Strengths**: AdapTS is lightweight, applicable to any FM, and shows significant performance improvement with low computational cost.
- **Weaknesses**: Hyperparameters are set a priori without tuning, which may not be optimal.

**Questions For Authors:**

- **Q1**: Why is $w_r$ a scalar, rather than being designed as a vector or a tensor?
- **Q2**: Many Test Time Adaption methods also meet the requirement of parameter freezing, so why don't the authors compare them? (such as https://doi.org/10.48550/arXiv.2501.04970)
- **Q3**: Why didn't the gradient - based approach work in experiments? In addition, gradient updates are not just about fine - tuning; for example, adding a small neural network to fit the residuals.

**Relation To Broader Scientific Literature:**

This paper focuses on TSFMs and clearly differentiates from previous time series continual learning work by avoiding direct gradient - based updates.

**Theoretical Claims:**

The paper correctly applies well - established theories such as the Woodbury matrix identity and exponential weighting. These theories firmly provide the basis for AdapTS to achieve efficient online adaptation.

---

> ### Author Rebuttal · Authors · 2025-03-31
>
> Dear reviewer, thank you for your kind words about our work and constructive comments/questions. We are particularly happy that you found that AdapTS gives "significant performance improvement with low computational cost" and is "applicable to any FM". We have included additional baseline comparisons against test-time adaptation methods and provide answers to your questions below.
>
> **Summary of comparisons against test-time adaptation and gradient-based methods**
>
> In response to your comments, we have now run experiments with several additional baselines: TAFAS (TTM-TAFAS); AdapTS-forecaster to predict residuals of the FM forecasts (TTM+Residual-Adjustment); a gradient-based online-ensembling approach for time series forecasting (OneNet-TTM) and fine-tuning the FM (TTM-Fine). The experiments are shown in the table below for TTM, the best performing FM in our experiments, on the ETT datasets. We give the results relative to the MASE performance of TTM+AdapTS, for example a score of 1.043 means that TAFAS performs 4.3\% worse than AdapTS.
> ||||Relative MASE to TTM+AdapTS||||
> |-|-|-|-|-|-|-|
> |**Dataset**|$H$|**TTM+TAFAS**|**TTM+Residual-Adjustment**| **OneNet-TTM**|**TTM-Fine**|
> |ETTh1|30|1.043|1.029|1.013|1.072|
> ||96| 1.038|1.021|1.277|1.092|
> ||336|1.028|1.011|1.284|1.092|
> |ETTh2|30|1.030|1.019|1.012|1.036|
> ||96|1.020|1.010|1.167|1.027|
> ||336|1.006|1.004|1.333|1.013|
> |ETTm1|30|1.033|1.065|1.905|1.031|
> ||96|1.051|1.059|1.285|1.062|
> ||336|1.049|1.050|1.576|1.059|
> |ETTm2|30|1.041|1.050|1.764|1.072|
> ||96|1.038|1.043|1.502|1.076|
> ||336|1.025|1.033|1.555|1.074|
>
> We find that, in all cases looked at, AdapTS performs better than the baselines and discuss each experiment in more detail in the sections below. We hope these experiments satisfy your comments on comparing to test-time adaptation and gradient-based methods and we believe those additions significantly strengthen the paper. We further note that we are currently running experiments across all the datasets and FMs and will add them to the paper once they complete.
>
> ****1. Why is $w_r$ a scalar, rather than being designed as a vector or a tensor?****
>
> If we understand correctly, when you mention $w_r$ you are referring to the weight used to merge the forecasts of the FM and AdapTS forecaster (in Eq 2.)? If so, making $w_t$ a vector would mean that there would be a different weight applied per time-step (using the element wise product between $w_r$ and $y_{t,FM}$). In the development stage of this work we did look at this, but found that using only per channel weights yielded better performance, due to overfitting. Therefore, we did not discuss it in the paper, however to clarify the text and to address your comment, we will add appropriate mentions of it in the paper. As for making the weights a tensor, we do not know exactly what you mean by this, could you please clarify?
>
> ****2. Comparison to test-time adaptation method TAFAS****
>
> We agree that this work would be greatly strengthened by comparisons with the test-time adaption baseline TAFAS. As mentioned before, results of these experiments and other baselines are given in the table at the start of our comment. We find that in all cases looked at AdapTS performs better than TAFAS. We also note that TAFAS requires backpropagating through the FM, making it computationally slower than AdapTS. This demonstrates the advantage of AdapTS over TAFAS.
>
> ****3. Why didn't the gradient-based approach work in experiments? In addition, gradient updates are not just about fine-tuning; for example, adding a small neural network to fit the residuals.****
>
> Gradient-based fine-tuning does not work well for the online learning of neural networks due to the problems studied in the continual learning literature. These problems mainly manifest as catastrophic forgetting whereby updating on new data the model forgets large amounts of information about old data. Hence, the gradient-based fine-tuning of TTM did not work for online updating (as shown in the table above). We note here that by using a linear model as the AdapTS-forecaster we can sidestep all the problems of continual learning which only occur for more complex models, while being significantly faster.
>
> Additionally, we have run an experiment in which we use the AdapTS-forecaster to predict residuals of the FM's forecasts and use this to modify the forecast. We present the results for this method in the table at the start of our comment, and find that it performs worse than AdapTS in all cases looked at.
>
> ****
>
> We thank the reviewer again for your constructive feedback. We hope given our answers and especially the added experiments comparing to TAFAS, fine-tuning, residual predictions and OneNet, we have satisfactorily addressed your concerns and believe that your comments have made our submission stronger.

---

> > ### Comment · Reviewer_6Y6Z · 2025-04-02
> >
> > Thank you for addressing my concerns during the author response period. I have updated my rating to reflect your clarifications.

---

> > > ### Author Response · Authors · 2025-04-08
> > >
> > > Thank you for your response and for updating your review based on our answers to your constructive comments! To demonstrate progress in our promise to extend the results given to you and other reviewers across more datasets and Foundation Models, we below present tables of additional experiments that have now finished running. Results in the first table are averaged over prediction length to save space. There remain some experiments which still have not finished and others we present in our reply to reviewer 26Ko.
> > >
> > > **Results of Comparisons to Online Adaptation Methods:**
> > >
> > > |||Relative MASE to FM+AdapTS|||||||||
> > > |---|---|---|---|---|---|---|---|---|---|---|
> > > |**Dataset**|**TTM+TAFAS**|**TimesFM+TAFAS**|**OneNet-TTM**|**OneNet**|**FSNet**|**TTM+RA** | **TimesFM+RA** | **VisionTS+RA** | **Chronos+RA** | **Moirai+RA** |
> > > |ETTh1|1.036|1.023|1.020|1.191|1.085|1.023|1.042|1.036|1.058|1.079|
> > > |ETTh2|1.019|1.011|1.011|1.171|1.025|1.013|1.015|1.020|1.018|1.028|
> > > |ETTm1|1.044|1.088|1.058|1.589|1.051|1.058|1.089|1.092|1.141|1.184|
> > > |ETTm2|1.035|1.057|1.042|1.607|1.074|1.044|1.073|1.102|1.098|1.130|
> > > |USWeather|1.021|1.051|1.336|1.376|1.514|1.037|1.062|1.079|1.102|1.069|
> > > |Weather|1.033|1.046|2.552|2.435|3.073|1.037|1.041|1.183|1.257|1.228|
> > > |Solar|1.029|1.077|1.898|1.762|1.990|1.048|1.090|1.052|1.044|1.138|
> > > |ECL|1.081|-|1.550|1.497|1.494|1.086|  -  |1.123|1.056|1.257|
> > > |Traffic|1.090|-|1.309|1.709|1.644|1.098|  -  |1.154|1.077|1.052|
> > >
> > >
> > > **Results on Per Second Datasets:**
> > >
> > > |Dataset|H|TTM|+AdapTS|TimesFM|+AdapTS|VisionTS|+AdapTS|Moirai|+AdapTS|
> > > |-|-|-|-|-|-|-|-|-|-|
> > > |SWind|30|0.728|-0.041|0.774|-0.089|1.492|-0.794|0.757|-0.07|
> > > ||96|1.77|-0.036|1.851|-0.111|2.299|-0.552|1.821|-0.082|
> > > ||336|4.759|-0.058|4.923|-0.208|4.975|-0.26|4.786|-0.079|
> > > |SSolar|30|0.286|-0.05|0.246|-0.016|0.738|-0.499|0.346|-0.109|
> > > ||96|0.581|-0.103|0.508|-0.03|0.907|-0.417|0.666|-0.183|
> > > ||336|1.543|-0.165|1.528|-0.165|1.647|-0.236|1.684|-0.229|
> > > |SCloud|30|0.94|-0.128|0.879|-0.079|0.98|-0.161|0.947|-0.132|
> > > ||96|1.06|-0.119|1.01|-0.085|1.075|-0.026|1.082|-0.137|
> > > ||336|1.285|-0.126|1.234|-0.094|1.24|-0.082|1.379|-0.212|
> > >
> > > We are very grateful for your constructive criticism throughout this process, which we believe has significantly strengthened our submission.

---

### Official Review · Reviewer_26Ko · 2025-03-12

**Overall Recommendation:** 4

**Summary:**

The paper proposes a method to combine foundation model forecasts with forecasts from an online learner. They innovate on 2 components, the online learner, and the algorithm to combine the forecasts. The online learner is a linear model in the frequency domain, learned via efficient closed form updates. The algorithm to combine the forecasts is based on the exponential weighting algorithm, combined with the idea of slow and fast learning. Experiments are performed on some standard datasets found in the literature, and shows  consistent improvement across 5 foundation models.

## update after rebuttal
I increased my rating as my concerns were addressed.

**Claims And Evidence:**

One major issue I have is that the paper claims to propose a method for adaptation of foundation models in the online/continual learning setting. However, the proposed method is actually an ensembling method - the foundation models are used as is, and are not adapted, instead, their predictions are ensembles with another online learning forecaster. I strongly encourage the authors to rename/reframe the proposed method, with the terminology of ensembling or exponential weights.

Unfortunately, Table 5 in the appendix makes the evidence for the idea of slow and fast weighting to be weakened, as the fast weighting only seems to match the full approach in most cases.

**Essential References Not Discussed:**

None that I'm aware of.

**Experimental Designs Or Analyses:**

I want to verify that the loss used in exponential weighting is from previous time steps, i.e. no data leakage?

**Methods And Evaluation Criteria:**

### Method
The paper is lacking details and formal notation. Especially in the methods section, AdaptTS-Forecaster should be described with more precise formal notation. I do not have a clear and exact understanding of how the linear model is learned in Fourier space. Information like what is the dimensionality of the weights, X, and Y, are missing.

### Evaluation
No baselines are presented. We cannot judge how well the proposed innovations perform because results are only presented for the proposed method. For the AdaptTS-Forecaster, relevant baselines would be to replace it with a similar linear model but with various forms of online gradient descent, and as well as FSNet (Pham et al, 2022). For the AdaptTS-Weighter, the relevant baselines would be the naive exponential weights algorithm (I suppose this is equivalent to slow weighter only, with results present in the appendix averaged over all models), and other similar methods from the online learning literature, e.g. hedge.

I encourage authors to present these results for each model, unlike table 5 - the results shouldn't really be averaged across models. Instead, it would be better to normalize the metrics to a baseline, then aggregate over the different datasets and settings (see the normalized MAE/MASE/... metrics in the Moirai and Chronos papers). That way, more results across different models and baselines can make it into the main paper, then the appendix contains the full breakdown.

**Other Comments Or Suggestions:**

None.

**Other Strengths And Weaknesses:**

None.

**Questions For Authors:**

None.

**Relation To Broader Scientific Literature:**

The contributions of the paper are novel and present ideas on how to ensemble predictions in an online fashion. I'm not too convinced about the need for time series foundation models for this case - they could be replaced with standard deep learning forecasting models, and the paper still reads exactly the same.

**Theoretical Claims:**

No theoretical claims were made.

---

> ### Author Rebuttal · Authors · 2025-03-31
>
> Dear Reviewer, thank you for your comments and constructive feedback. We are especially happy you found that "the contributions of the paper are novel and present ideas on how to ensemble predictions in an online fashion". We answer the comments you had below and are thankful that you pointed out issues in notation and framing which we have fixed.
>
> **Ensembling: You identify that the proposed method is actually an ensembling method - encouraging us to reframe with the terminology of ensembling or exponential weights.**
>
> We have edited the paper to incorporate this feedback and to frame the method in ensembling terminology. Additionally, to fully take your feedback into account we have renamed our method to **ELF**, standing for **E**nsembled with online **L**inear **F**orecaster. e.g. TTM+AdapTS becomes TTM+ELF. To reduce confusion in the rebuttal we have resorted to using the name AdapTS.
>
> **Details and Formal Notation: You point out that the paper is lacking details and formal notation in places. In particular, the AdapTS-Forecaster description could be made clearer.**
>
> Thank you for pointing this out; we feel that addressing these points has improved the quality of the paper. We have now updated the paper to make the explanation of the AdapTS-Forecaster clearer, clarifying the notation and providing a more precise explanation of model fitting. We have added dimensionality information where missing, as well. Additionally, an algorithmic description of fitting the AdapTS-Forecaster is now given in the Appendix.
>
> **Baselines: You identify a lack of comparison with relevant baselines, both for the AdapTS-Forecaster and AdapTS-Weighter**
>
> Thank you for this comment, it was mirrored in the feedback of other reviewers as well and has now been addressed. We have compared the AdapTS-Forecaster with FSNet, FITS and a linear model trained by online gradient descent (OGD) and we have compared the AdapTS-Weighter with Hedge as recommended. The results for TTM on the ETT datasets are provided in the table below. Results are given relative to our approach so that values above 1 correspond to worse performance. In each case results are worse than using AdapTS. These results will be extended to all datasets and FMs.
>
> ||||Relative MASE to TTM+AdapTS|||
> |-|-|-|-|-|-|
> |**Dataset**|$H$|**Hedge-Weighter**|**OGD-Forecaster**|**FSNet-Forecaster**|**FITS-Forecaster**|
> |ETTh1|30|1.020|1.037|1.029|1.037|
> ||96|1.018|1.025|1.023|1.026|
> ||336|1.013|1.007|1.027|1.013|
> |ETTh2|30|1.017|1.023|1.021|1.026|
> ||96|1.008|1.010|1.008|1.010|
> ||336|1.003|1.004|1.009|1.003|
> |ETTm1|30|1.021|1.070|1.068|1.070|
> ||96|1.013|1.061|1.060|1.059|
> ||336|1.009|1.057|1.057|1.054|
> |ETTm2|30|1.018|1.051|1.051|1.052|
> ||96|1.015|1.040|1.042|1.041|
> ||336|1.012|1.034|1.034|1.032|
>
> **Results Presentation: You propose that it would be better to normalize the metrics to a baseline, then aggregate over the different datasets and settings.**
>
> Thank you for suggesting this way to incorporate model results into the main text. We have updated the paper to normalized MASE (w.r.t. the naive seasonal forecaster) and once all our additional experiments have run we will aggregate in the way you propose to ensure that as many of them can be presented in the main text as possible.
>
> **I want to verify that the loss used in exponential weighting is from previous time steps, i.e. no data leakage?**
>
> Yes, the true values used to train both the weighter and the forecaster at some time step _t_ are only those already seen by that time step (i.e. 0 to t).
>
> **"Time series foundation models could be replaced with standard deep learning forecasting models, and the paper still reads exactly the same."**
>
> You are correct that this method could in principle be applied to standard deep-learning forecasters. We decided to focus on FMs due to the particular difficulties and computational cost of finetuning these approaches online.
>
> ## Alteration Summary
> Thank you again for your feedback. Below is a summary of improvements made based upon your feedback. We look forward to hearing your thoughts.
>
> 1) Compared Forecaster and Weighter against Baselines
> 2) Addressed issues with notation and terminology
> 3) Reframed our approach with the terminology of ensembling
> 4) Updated to Normalized MASE

---

> > ### Comment · Reviewer_26Ko · 2025-04-08
> >
> > Thanks for the detailed rebuttal authors, I have updated my rating as my concerns have been addressed. I look forward to reading the updated version of the paper.
> >
> > Although there's one more thing, is there any response on this statement from my original review?
> >
> > "Unfortunately, Table 5 in the appendix makes the evidence for the idea of slow and fast weighting to be weakened, as the fast weighting only seems to match the full approach in most cases."

---

> > > ### Author Response · Authors · 2025-04-08
> > >
> > > Dear Reviewer, thank you for your response and for updating your review. We are glad the rebuttal addressed your concerns. To demonstrate that we are continuing to complete the suggested experiments, we present below the AdapTS-Weighter and Forecaster baselines for the TTM FM extended to all datasets and, to reduce size, averaged over prediction length (other experiments remain in progress).
> > >
> > > **Results of Replacing the AdapTS-weighter or AdapTS-forecaster with Different Methods:**
> > >
> > > |||Relative MASE to TTM+AdapTS|||
> > > |---|---|---|---|---|
> > > | **Dataset** | **Hedge-Weighter** | **OGD-Forecaster** | **FSNet-Forecaster** | **FITS-Forecaster** |
> > > | ETTh1       |1.017|1.023|1.026|1.025|
> > > | ETTh2       |1.009|1.012|1.013|1.013|
> > > | ETTm1       |1.014|1.063|1.062|1.061|
> > > | ETTm2       |1.015|1.042|1.042|1.042|
> > > | US Weather  |1.012|1.039|1.026|1.039|
> > > | Weather     |1.011|1.037|1.038|1.037|
> > > | Solar       |1.024|1.041|1.042|1.056|
> > > | ECL         |1.012|1.091|1.095|1.102|
> > > | Traffic     |1.011|1.098|1.065|1.103|
> > >
> > > To answer your question about the AdapTS-weighter ablation presented in Table 5, you are correct that in many cases using the AdapTS-weighter leads to the same performance as the fast weigher. However, we still believe the combination of the the fast and slow weighers to a valuable contribution for the following reasons: a) for the more complex datasets (ECL and Traffic) using the AdapTS-weighter leads to better results than using the fast weighter; b) for the rest of the datasets it does not lead to any degradation of performance compared to using the fast weigher; c) It has a minimal computation overhead. We will make further clarifications to the paper, based on this comment.
> > >
> > > Once again, thank you greatly for the constructive feedback! In our eyes it has certainly strengthened the work.

---

### Decision · Program_Chairs · 2025-05-01

**Decision:**

Accept (poster)

**Comment:**

This paper introduces a simple method to improve time series foundation model forecasts using online feedback. Reviewers found the approach intuitive and lightweight, with clear empirical gains across several models. While initial concerns were raised about framing, baseline comparisons, and clarity, the authors provided strong rebuttals, reframed the method appropriately, and added relevant baselines including test-time adaptation approaches. The method is general, efficient, and shows consistent benefits, justifying acceptance.  The authors are advised to factor in the reviewer feedback when preparing the updated version of the paper, including expanding related work.